# GUST1.0: A GPU-accelerated 3D Urban Surface Temperature Model

Shuo-Jun Mei[1,2*], Guanwen Chen[1,2], Jian Hang[1,2], Ting Sun[3]

[1] School of Atmospheric Sciences, Sun Yat-sen University, and Southern Marine Science and Engineering Guangdong Laboratory (Zhuhai), Zhuhai 519082, PR China

[2] China Meteorological Administration Xiong'an Atmospheric Boundary Layer Key Laboratory, Xiong'an, P.R. China

[3] Department of Risk and Disaster Reduction, University College London, London, UK

*Correspondence to*: Shuo-Jun Mei (meishj@mail.sysu.edu.cn)

## Abstract

The escalating urban heat, driven by climate change and urbanization, poses significant threats to residents' health and urban climate resilience. The coupled radiative-convective-conductive heat transfer across complex urban geometries makes it challenging to identify the primary causes of urban heat and develop mitigation strategies. To address this challenge, we develop a GPU-accelerated Urban Surface Temperature model (GUST) through CUDA architecture. To simulate the complex radiative exchanges and coupled heat transfer processes, we adopt Monte Carlo method, leveraging GPUs to overcome its computational intensity while retaining its high accuracy. Radiative exchanges are resolved using a reverse ray tracing algorithm, while the conduction-radiation-convection mechanism is addressed through a random walking algorithm. The validation is carried out using the Scaled Outdoor Measurement of Urban Climate and Health (SOMUCH) experiment, which features a wide range of urban densities and offers high spatial and temporal resolution. This model exhibits notable accuracy in simulating urban surface temperatures and their temporal variations across different building densities. Analysis of the surface energy balance reveals that longwave radiative exchanges between urban surfaces significantly influence model accuracy, whereas convective heat transfer has a lesser impact. To demonstrate the applicability of GUST, it is employed to model transient surface temperature distributions at complex geometries on a neighborhood scale. Leveraging the high computational efficiency of GPU, the simulation traces $10^5$ rays across $2.3\times10^4$ surface elements in each time step, ensuring both accuracy and high-resolution results for urban surface temperature modeling.

## 1. Introduction

Urban overheating has become a pressing issue due to the combination effects of global warming, heatwaves, and rapid urbanization (Feng et al., 2023). The Urban Heat Island (UHI) effect is characterized by higher surface and air temperatures in urban areas than in surrounding rural areas, which exacerbates the urban overheating (Manoli et al., 2019). It is estimated that more than 1.7 billion people and 13,000 cities are facing urban overheating problems (Tuholske et al., 2021). Exposure to extreme urban heat poses a significant threat to residents' health, contributing to increased mortality and morbidity (Ebi et al., 2021).

To tackle urban overheating, a precise understanding of the factors driving excessive surface heat is essential, making accurate modeling of urban surface temperatures a critical step toward developing effective mitigation strategies. Urban surface temperatures are commonly simulated with urban land surface schemes (LSMs). To capture the complex exchanges of energy and momentum within an urban environment, these schemes range from simplified approaches that represent the city as a single impervious slab to advanced frameworks that explicitly incorporate the three-dimensional geometry of buildings with varying heights and material properties. The Urban-PLUMBER project has evaluated 32 such schemes (Grimmond et al., 2010; Grimmond et al., 2011), and classified them into ten categories based on the level of three-dimensional detail represented. The most detailed of these are the building-resolved schemes, which explicitly solve airflow and heat transfer while representing the full three-dimensional urban landscape.

Building-resolved models, such as VTUF (Nice, 2016) and computational fluid dynamics (CFD) tools (Carmeliet and Derome, 2024), solve the governing physical processes at high spatial and temporal resolution. These models are powerful tools for examining the urban thermal balance and identifying the primary drivers of urban heat (Carmeliet and Derome, 2024). They enable a quantitative evaluation of the contribution of each process, such as conduction, radiation, and convection, to the overall thermal balance. This is particularly important for Asian cities, which are characterized by high-density, high-rise developments and complex urban geometry. Findings from the Scaled Outdoor Measurement of Urban Climate and Health (SOMUCH) project highlight the intricate influence of building morphology

on the thermal environment, especially under super-high-density conditions (Hang and Chen, 2022).
These effects arise from complex three-dimensional urban landscapes, including irregular building forms
and intricate shading patterns. Accordingly, models representing high-density Asian cities need greater
accuracy and flexibility to account for these features.
Building-resolved urban surface temperatures are determined by the coupled heat transfer processes of
conduction, radiation, and convection (Krayenhoff and Voogt, 2007). These heat transfer processes in
urban areas differ from those in rural areas. First, urban materials typically have a lower heat capacity,
allowing them to heat up more quickly and reach higher temperatures (Wang et al., 2018). Secondly, the
complex three-dimensional geometry of urban environments leads to multiple reflections, which enhance
the absorption of solar radiation by surfaces and reduce the net reflected radiation escaping to the
atmosphere (Yang and Li, 2015). Thirdly, the densely packed buildings weaken the urban wind and thus
reduce the convective transfer and further limit the heat loss (Wang et al., 2021).
A well-designed building-resolved model needs to accurately capture these heat transfer processes. Table
1 summarizes the models for urban surface temperatures and their schemes for conduction, radiation,
and convection. For heat conduction, 1D models are commonly used due to the relatively thin walls of
buildings in urban areas. For convective heat transfer, both parameterized convective coefficients and
CFD simulations are commonly used. CFD simulations can better capture the spatial variations in air
temperature in densely built urban areas, but the computational cost is much higher.
The key distinction among these models lies in their radiation schemes, as radiation is the primary energy
input into the thermal system of urban surfaces. Moreover, simulating complex urban radiative transfer
requires significant computational resources, necessitating simplifications and parameterizations to make
the simulation more applicable. For the radiative exchange between urban surfaces, the radiosity method
is widely adopted. This approach first collects luminous energy from direct solar and diffuse sky sources
and then redistributes reflected energy according to view factors, which quantify the geometric
relationships among surfaces. View factors can be determined analytically for simple geometries,
estimated with the discrete transfer method (hemisphere discretization and ray counting), or calculated
using Monte Carlo ray tracing (MCRT). However, the radiosity method assumes purely diffuse
reflections and depends on precise view-factor calculations, making it less accurate for complex urban
geometries and surfaces containing semi-transparent materials.
In contrast, the MCRT approach offers greater flexibility and has been widely employed to model solar
radiation on complex urban surfaces (Kondo et al., 2001). More recently, its use has expanded beyond
radiative transfer to encompass coupled conduction, convection, and radiation processes (Villefranque et
al., 2022). In backward MCRT, the energy of the incident light is divided into a large number of photons.
By tracking the path of these photons and counting the number of photons absorbed, the net solar
radiation reaching a given surface can be calculated. For example, the HTRDR-Urban adopted the
backward MCRT, to calculate the solar radiation considering multiple reflections (Schoetter et al., 2023).
Building on this concept, Tregan et al. (2023) proposed a theoretical framework to solve linearized
transient conduction-radiation problems with Robin's boundary condition in complex 3D urban geometry.
Based on that framework, Caliot et al. (2024) developed a probabilistic model to simulate urban surface
temperatures, using ray-tracing, walk-on-sphere and double randomization techniques. Their model
leverages advancements in computer graphics for image synthesis and the MCM, enabling it to
effectively handle large and complex 3D geometries.
The MCRT method has demonstrated strong capability for accurately modeling coupled heat and
radiation processes in complex urban environments, but its high computational cost and low efficiency
currently limit its application to real-world urban configurations. Although several models listed in Table
1 have been validated against field measurements, others remain unverified and rely on various
assumptions and parameterizations, which reduces confidence in their accuracy. Furthermore, the use of
field measurement data for model validation faces persistent challenges: 1) limited test points due to
regulatory constraints and installation difficulties, 2) uncertainty in infrared imagery caused by varying
view angles, and 3) heterogeneity in the optical and thermal properties of building materials.
This study aims to develop a GPU-accelerated Urban Surface Temperature (GUST) model to enhance
the computational speed of Monte Carlo Method. The model is designed to operate at the neighborhood
scale and to capture microscale processes, including complex shading patterns, multiple reflections of
solar radiation, and longwave radiative exchanges between building surfaces and the ground. The
ultimate objective is to identify the physical drivers of extreme heat in high-density urban neighborhoods.
The absorption and reflection of longwave and solar radiation on outdoor surfaces modeled using the
reverse Monte Carlo ray tracing (rMCRT) algorithm. The resulting solar and longwave radiation are then
treated as heat flux boundary conditions for the 1D heat conduction model, which employs the Monte
Carlo random walk method to calculate surface temperatures. High spatial-temporal resolution surface
temperature data from a scaled measurement (SOMUCH) is employed to validate the parameterization
and assumptions in this model.
The paper is organized as follows. Sect. 2 outlines the model structure and describes the algorithms used
for the submodels. Sect. 3 presents the validation and evaluation of the model by comparing it with
experimental data. Sect. 4 includes an example demonstrating how the model can be applied to complex
geometries. Sect. 5 discusses the applications, limitations, and future development of the model. Lastly,
Sect. 6 provides the conclusions.

**Table 1**. Overview of building-resolved models for urban surface temperature. The view factors are
solved by both DTM (Discrete transfer method), analytical model, and Monte Carlo ray tracing method.

| Model | Solar Irradiation | Reflections and longwave exchange | Conduction | Convection | Validation |
|---|---|---|---|---|---|
| **HTRDR-Urban** (Schoetter et al., 2023) | Backward Monte Carlo ray tracing | Backward Monte Carlo ray tracing | Monte Carlo random walking | Parameterized | N.A. |
| **MUST** (Yang and Li, 2013) | Sunlit-shaded distributions | Radiosity Method, DTM view factors | 1D heat conduction | Parameterized | Thermal scanner and IRT (Voogt and Oke, 1998) |
| **TUF-3D** (Krayenhoff and Voogt, 2007) | Sunlit-shaded distributions | Radiosity Method, analytical view factors | 1D heat conduction | Parameterized | Thermal scanner and IRT (Voogt and Oke, 1998) |
| **SOLENE Microclimat** (Imbert et al., 2018) | Sunlit-shaded distributions. | Radiosity Method, analytical view factors | 1D heat conduction | Coupling CFD simulation | Thermographies measurement (Hénon et al., 2012) |
| **Envi-Met** (Eingrüber et al., 2024) | Flux reduction coefficients | Radiosity Method, DTM view factors | 1D heat conduction | Coupling CFD simulation | Field measurements (Forouzandeh, 2021) |
| **uDALES** (Owens et al., 2024) | Sunlit-shaded distributions | Radiosity Method, DTM view factors | 1D heat conduction | Coupling CFD simulation | N.A. |
| **PALM** (Resler et al., 2017) | Sunlit-shaded distributions | Radiosity Method, Analytical and DTM view factors | Empirical heat conductivity | Coupling CFD simulation | Field measurement (Resler et al., 2017) |
| **MITRAS** (Salim et al., 2018) | Meso-scale radiation scheme | Meso-scale radiation scheme (METRAS) | Force-restore method | Coupling CFD simulation | N.A. |
| **OpenFOAM** (Rodriguez et al., 2024) | Sunlit-shaded distributions | Radiosity Method, DTM view factor | 1D heat-moisture diffusion. | Coupling CFD simulation | N.A. |
| **FLUENT** (Toparlar et al., 2015) | Sunlit-shaded distributions | Radiosity Method, DTM view factor | Shell conduction | Coupling CFD simulation | Field measurement (Toparlar et al., 2015) |


## 2. Model design

The main objective of GUST is to resolve the coupled radiative–convective–conductive heat transfer processes occurring across complex urban geometries. These coupled processes represent one of the core physical mechanisms driving the urban heat island effects (Manoli et al., 2019). The model is developed based on reduced-scale outdoor measurements conducted within a simplified urban environment (Hang and Chen, 2022). In this experimental setup, complex glazing systems and green infrastructure are intentionally excluded to isolate and validate the core radiative–convective–conductive heat transfer mechanisms. GUST uses a time-dependent heat conduction model to couple radiative, convective, and conductive heat transfer processes, as illustrated in Fig. 1.

The convective and radiative heat transfer at urban surfaces is treated as boundary conditions for the 1D heat conduction model. For the outdoor side, the heat flux ($q_{out}$) is the sum of radiative (longwave $q_l$ and solar $q_s$) and convective heat flux ($q_{c,out}$).

$$q_{out} = q_l + q_s + q_{c,out} \qquad (1)$$

The absorbed solar radiation, $q_s$ is the sum of direct solar irradiation ($q_{s,o}$) and diffuse solar irradiation ($q_{s,r}$), expressed by: $q_s = q_{s,o} + q_{s,r}$. The longwave radiation flux $q_l$ includes the radiation between urban surfaces ($q_{l,urban}$) and between urban surfaces and the sky ($q_{l,sky}$), represented as $q_l = q_{l,urban} + q_{l,sky}$.

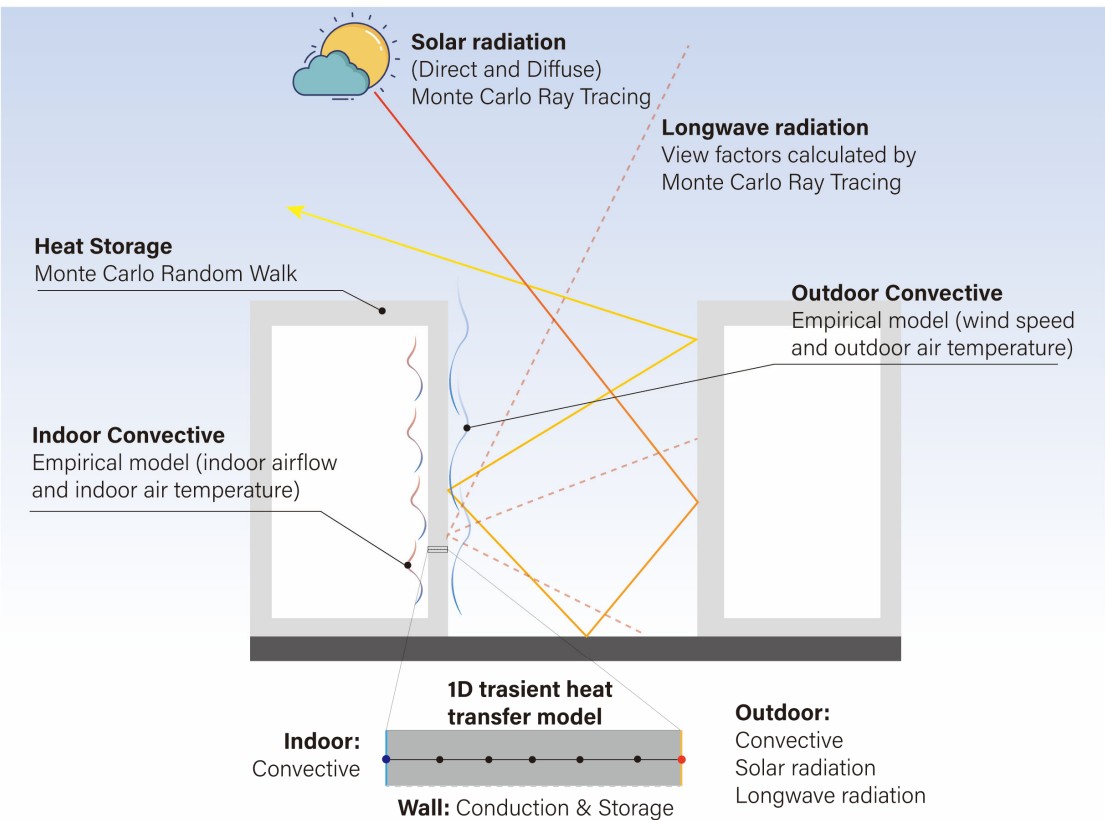

142

**Figure 1: The model design of GUST. In this model, 1D transient conductive heat transfer is considered for urban surfaces the system (e.g., walls, roofs, and ground). They are composed of multiple layers where the thermal properties are uniform and isotropic. All urban surfaces are assumed to be opaque in this study.**

In this model, all urban surfaces are represented as triangular facets in STL format, with each triangular facet treated as a single element. Ray tracing and heat-conduction calculations are performed at the centroid of each element. The spatial resolution of the simulation can be refined by using smaller triangular facets, thereby increasing the number of elements. Fig. 6 illustrates the triangulated representation of the urban surfaces.

## 2.1. Conduction sub-model

The Monte Carlo random walking method is used to solve the 1D heat conduction (Talebi et al., 2017). Compared to finite volume method, this approach is insensitive to the complexity of urban geometry and boundary conditions (Villefranque et al., 2022; Caliot et al., 2024). In the present version, the heat conduction along the wall span is neglected. The one-dimensional (1D) transient heat conduction equation is:

$$\frac{\partial}{\partial t}T = \alpha \frac{\partial^2 T}{\partial x^2} \tag{2}$$

where $\alpha = \frac{k}{\rho c_p}$ is the solid thermal diffusivity and $k$ the thermal conductivity, $\rho$ the density, $c_p$ the
specific heat capacity. The ground, walls and roofs are composed of multiple layers. In the Monte Carlo
random walking method, the heat conduction equation is replaced by finite difference approximation as:
$$T(x, t + \Delta t) = P_t T(x, t) + P_{x-}T(x - \Delta x, t + \Delta t) + P_{x+}T(x + \Delta x, t + \Delta t) \tag{3}$$

where $P_t = \frac{1}{1+2Fo}$ is defined as probability of time step; $P_{x-} = P_{x+} = \frac{Fo}{1+2Fo}$. where $P_{x-}$ and $P_{x+}$
respectively represent the probabilities of stepping to the points $(x - \Delta x, t)$ and $(x + \Delta x, t)$. Here,
$Fo = \frac{k\Delta t}{\rho c_p (\Delta x)^2}$ These coefficients are nonnegative probabilistic values and
$$P_t + P_{x-} + P_{x+} = 0 \tag{4}$$

The Monte Carlo random walking algorithm is schematically illustrated in Fig. 2. The core idea is that
particles walk by following rules:
1) Start a random walk at point $x$.
2) Generating a random number (R) between 0 and 1.
3) Determine walking direction by conditions
$$\begin{cases} 0 < R < P_{x-}: & x \rightarrow (x - \Delta x) \\ P_{x-} < R < (P_{x-} + P_{x+}): x \rightarrow (x - \Delta x) \\ (P_{x-} + P_{x+}) < R: & x \rightarrow (x), T(i) = T(i) + T(x, t - \Delta t) \end{cases} \tag{5}$$

4) If the next point is not on the boundary repeat step 2 and 3 and if it is on the boundary, record $T(i) =$
$T(i) + T$ at the boundary and go to step 1.
5) After $N$ random walking, temperature at point $x$ is calculated by
$$T(x) = \frac{T(i)}{N} \tag{6}$$

When a particle reaches a heat flux, convective or interface boundary, its movement follows the following
rules.
1) Heat flux boundary
When the particle walks to the boundary of heat flux ($q$), it is bounced back and record the temperature
$T_{hf}$, which is calculate by $T_{hf} = \frac{q\Delta x}{k} + \frac{q}{2k}(\Delta x)^2$.
2) Convective boundary
The heat flux of a convective boundary is calculated by $q = h(T_w - T_a)$, where h is the heat transfer
coefficient and $T_w$ the wall temperature and $T_a$ the air temperature. The wall temperature is calculated
by
$$T_w = \frac{1}{1 + Bi}T(x - \Delta x) + \frac{Bi}{1 + Bi}T_a \tag{7}$$

Where $P_x = \frac{1}{1+Bi}$, $P_a = \frac{Bi}{1+Bi}$, $Bi = \frac{h\Delta x}{k}$. When the particle reaches the convective boundary, a new
random number R was generated and moves as follows:
$$\begin{cases} 0 < R < P_x: & \rightarrow \text{bounced back} \\ P_x < R < 1: & \rightarrow \text{absorbed by air with T(i) = T(i) + } T_a \end{cases} \tag{8}$$

3) Interface between two layers
The interface between layers is flux continuity, i.e. the conductive fluxes are equal on both sides of the
interface. The heat conductivities on left and right sides of the interface are $k_A$ and $k_B$. The conductive
heat fluxes on both sides are equal, i.e., $-k_A\frac{dT}{dx} = -k_B\frac{dT}{dx}$. When a particle reaches the interface, it may
be reflected or move to the next layer. A new random number $R$ is generated. The particle moves by
following
$$\begin{cases} 0 < R < P_{x-}: & \rightarrow \text{bounced back to layer A} \\ P_{x-} < R < 1: & \rightarrow \text{move to layer B} \end{cases} \tag{9}$$

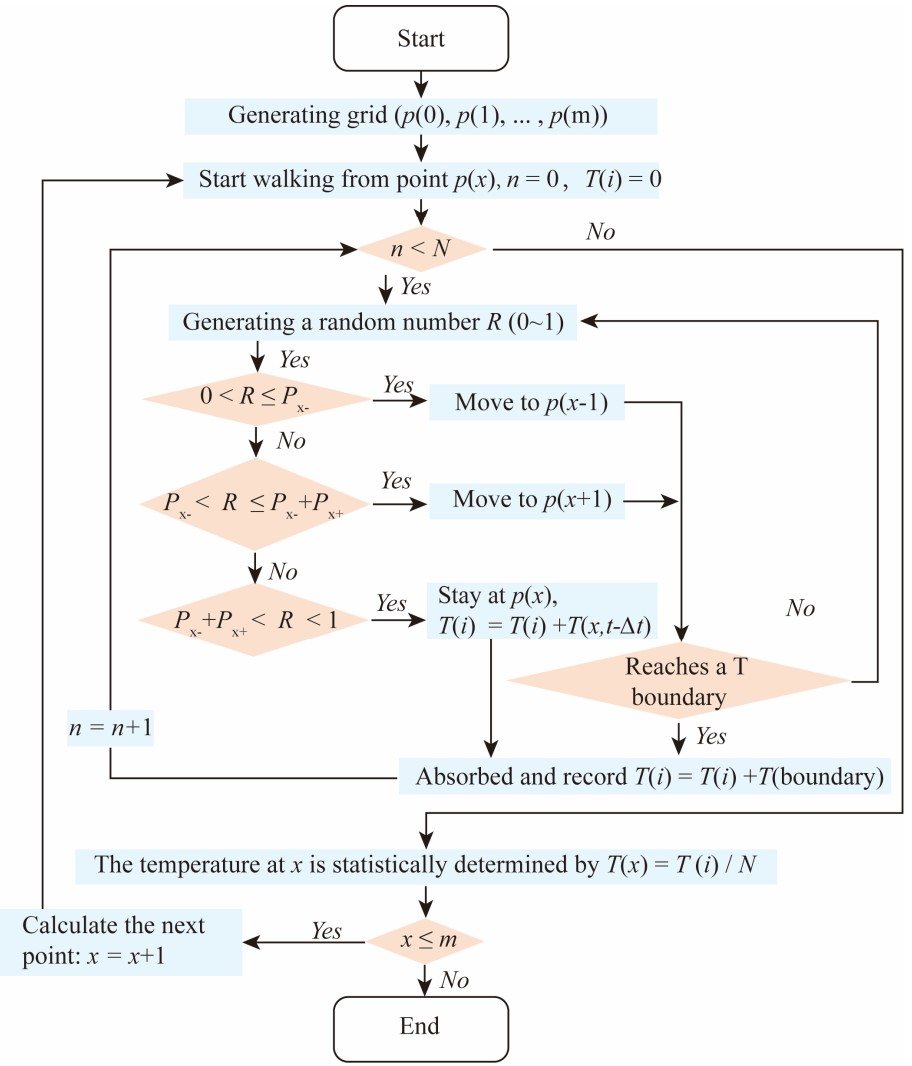

**Figure 2: Flowchart of the Monte Carlo random walking algorithm for 1D heat conduction. At each point, the particle movement stops after $N$ random walks. Each walk stops when particle either reaches a fixed temperature boundary or remains stationary. Orange diamonds indicate decision points with two possible outcomes (Yes/No).**

## 2.2. Solar radiation sub-model

The solar radiation $q_s$ is calculated on each triangular facet using the reverse Monte Carlo Ray Tracing (rMCRT) method, which inherently accounts for both shaded and sunlit areas. In the rMCRT, the ray starts from the target points, instead of starting from the sky or sun in the ray tracing method (Caliot et al., 2024). This method ensures that enough photons reach the target point to obtain a statistical result.

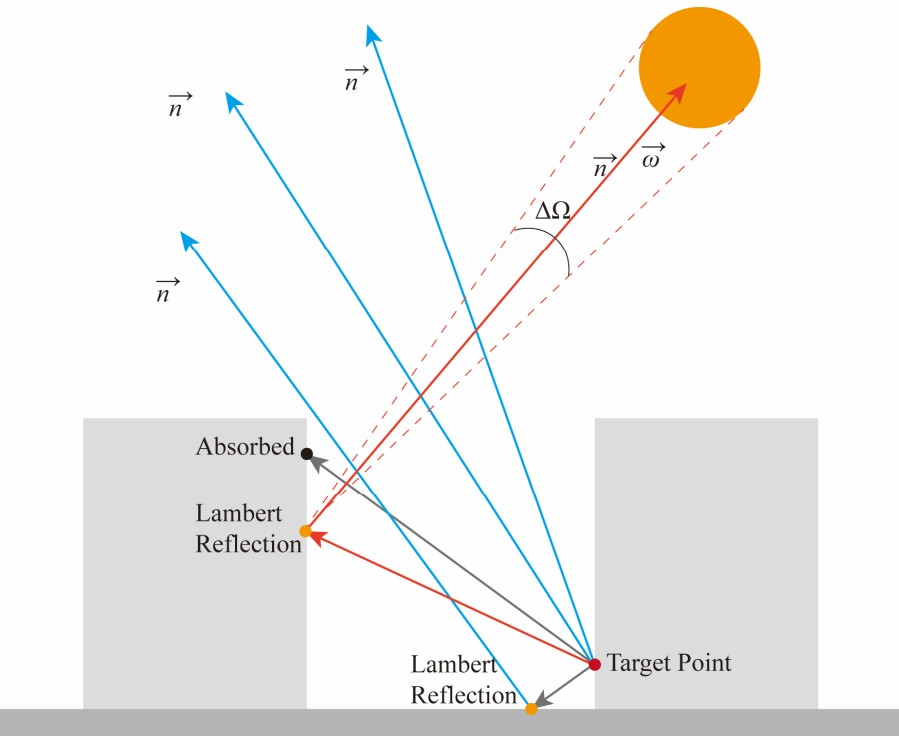

**Figure 3: Schematic illustration of the reverse MCM ray tracing method for calculating the direct and diffuse**
**solar radiation.**
The procedure of reverse MCRT is schematically explained in Fig. 3. In total, $N$ photons leave the target
point in random directions ($\vec{r}$), which is determined by the azimuth $\theta_a$ and incidence angle $\eta_a$. These
angles are calculated by $\theta_a = 2\pi R_1$ and $\eta_a = \arccos(1 - 2R_2)$, where $R_1$ and $R_2$ are random
numbers between 0 and 1.
When a photon reaches the surface, it can be absorbed or reflected via Lambert's law. To determine
whether this photon is absorbed, a random number $R_{ab}$ (ranging from $0 \sim 1$) is generated. When $R_{ab} >$
$\alpha_s$ (surface albedo), the photon is absorbed by the surface. When $R_{ab} < \alpha_s$, the photon is reflected. All
surfaces are considered Lambertian and the direction of reflect solar beam is determined by the azimuth
$\theta_a$ and incidence angle $\eta_a$ of that surface. At each reflection, $\theta_a$ and $\eta_a$ are recalculated by
regenerating new random numbers.
When the photon reaches the "sky" in the direction of $\vec{r}$, its angle ($\theta_{ns}$) with the reverse solar direction
$\overrightarrow{\omega_{sun}}$ is calculated. When $\theta_{ns} < \Delta\Omega_d$, that photon is marked as reaching the "Sun", otherwise, that
photon is marked as reaching the "Sky". The direct ($q_{s,o}$) and diffuse ($q_{s,r}$) solar radiation reaching the
target point can then be statistically determined by:

$$q_{s,o} = \frac{\pi I_{s,o}}{\Delta\Omega_d N} \sum_{\theta_{ns} < \Delta\Omega_d} \left| \vec{\omega}_{sun} \cdot \vec{n} \right| \qquad (10)$$

$$q_{s,r} = \sum_{\theta_n > d\Delta\Omega_d} \frac{I_{s,r}}{N} \qquad (11)$$

where $I_{s,o}$ and $I_{s,r}$ is the direct normal irradiance and diffuse solar radiation. The ratio between the
direct and diffuse solar radiation is calculated by the model proposed by (Reindl et al., 1990).
The rMCRT requires a large number of rays to achieve statistically reliable results. To accelerate the
simulation, the model is run in parallel on GPUs (Graphics Processing Units) using the CUDA® platform
(Yoshida et al., 2024). The advantage of GPUs is that they have a large number of cores, which enables
them to handle many parallel tasks simultaneously. GPUs are particularly well-suited for accelerating
MCRT, since each ray tracing operation is independent.
The GPU parallel computing is executed using two strategies, depending on the total number of elements.
As illustrated in Fig. 4, Strategy 1 calculates the radiative flux point by point, emitting $n$ photons for
ray tracing simulation. Each photon is processed in a separate GPU core. Once the ray tracing process is
complete, the results from the GPU cores are copied to the CPU, where radiative flux at each point is
calculated. Strategy 2 calculates the radiative flux for all points simultaneously, with each GPU core
computing the flux for a single point. The ray tracing of $n$ photons is performed iteratively on the GPU.
The advantage of Strategy 1 is the efficient utilization of GPU cores when the number of points and
elements is small. However, its disadvantage is that it requires a large amount of memory when the
number of points is large. In contrast, Strategy 2 requires significantly less memory and only transfers
data to the CPU once, making it highly efficient when the number of points and elements is large.

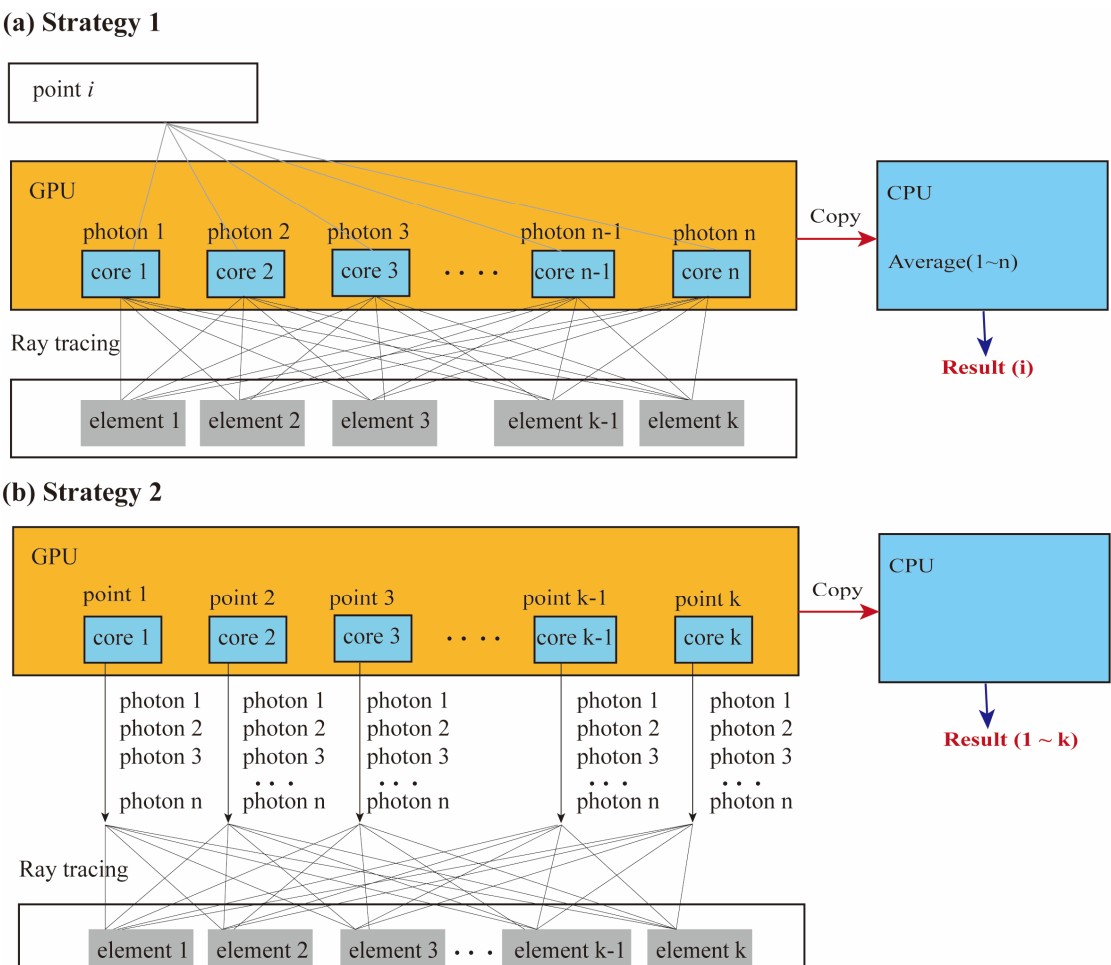

**(a) Strategy 1**

**(b) Strategy 2**


**Figure 4: Two strategies for GPU parallel computing. (a) The ray tracing is conducted point by point. For**
**each point, $n$ photons are emitted. Each GPU core calculates one photon. (b) The ray tracing is conducted**
**for all points at one time. Each GPU core calculates one point. The ray tracing of $n$ photons is performed**
**iteratively within the GPU core.**
The space angle of the Sun ($\Delta\Omega_d$) and the number of photons ($N$) can significantly affect the accuracy of
reverse MCM. To evaluate this influence, a series of test cases are conducted, in which the direct solar
radiation at a ground point is calculated. The solar radiation on the open ground can be calculated
theoretically, as there is no shading from buildings.
Figure 5 shows the errors of simulations using different values of $N$ and $\Delta\Omega_d$. The simulation time of
each case is also indicated in that figure. When the number of photons is increased from $N = 10^5$ to
$N = 10^7$, the simulation time increases from 0.05s to 1.15s, which is an increase of 23 times. The
relatively slow increase in simulation time is a result of the parallel computing capabilities of the GPU.
In each scenario, the model was run 20 times to observe the difference between each run.
A small $\Delta\Omega_d$ reduce the photon number reaching the Sun, thus increasing the error, where the $\Delta\Omega_d$ is
calculated from a 2D angle $\theta$ as $\Delta\Omega_d = 2\pi(1 - \cos(\theta))$. For example, the error in cases with $\theta = 3°$
greater than that in cases with $\theta = 6°$. A larger number of photons is needed to compensate for this error.
For example, the case with $\theta = 3°$ and $N = 10^7$ shows acceptable accuracy. However, the case with
$\theta = 6°$ shows a comparable accuracy when $N = 10^6$ and takes less simulation time.
In the subsequent simulations, $\theta = 6°$ and $N = 10^6$ are applied to balance accuracy and simulation
time.

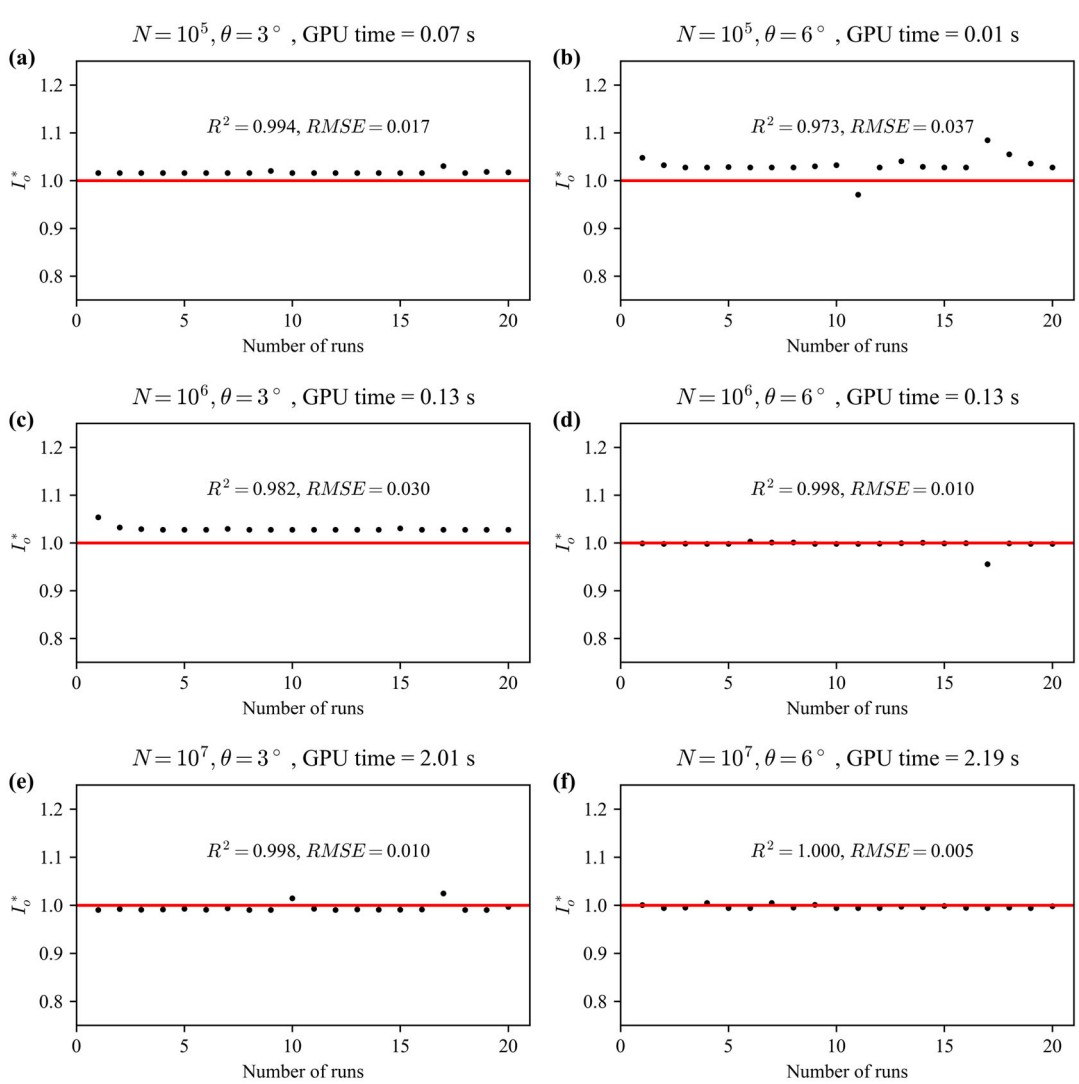


**Figure 5: Numerical errors of direct solar radiation estimation using Monte Carlo method. The simulated**
**solar radiation ($I_{o,sim}$) is normalized by the true value ($I_{o,true}$) and is expressed by ($I_o^* = \frac{I_{o,sim}}{I_{o,true}}$), where $I_o^* =$**
**$1.0$ represents an exact reproduction of the solar radiation. The test cases use different space angles of sun**
**$\Delta\Omega_d = 2\pi(1 - \cos(\theta))$ and photon numbers ($N$). The red lines represent the true value, and dots represent**
**the simulated data.**

## 2.3. Longwave radiation sub-model

The view factors between the surfaces, as well as from the surfaces to the sky, are also calculated using
the Monte Carlo ray tracing model, as illustrated in Fig. 6. The urban surfaces are divided into multiple
triangular elements $N_{ur}$. The view factor from element $S_i$ to element $S_j$, denoted as $F_{i,j}$, is calculated
by emitting $N$ photons from the centroid of element $S_i$. The algorithm then counts the number of
photons $n_{i,j}$ that reach element $S_j$. Finally, the view factor $F_{i,j}$ is calculated by $F_{i,j} = n_{i,j}/N$. The sky
view factor is also determined in this approach by treating the sky as an urban surface.
The longwave radiative heat exchange between the surfaces, as well as from the surfaces to the sky, is
calculated by:

$$q_l = F_{i,sky}\varepsilon(R_{l.in} - \sigma T_i^4) + \varepsilon\sigma \sum_{j=1}^{j=N_{ur}} F_{i,j}\left(T_j^4 - T_i^4\right) \tag{12}$$

where $\varepsilon$ is the material emissivity, $\sigma$ is Stefan–Boltzmann constant ($= 5.67 \times 10^{-8}$) (W m$^{-2}$ K$^{-1}$), $R_{l.in}$ is
the downward longwave radiation from the sky, $F_{i,sky}$ is the sky view factor of element $S_i$. The surface
temperature from the previous step ($T_i$ and $T_j$) is used to calculate the longwave radiative heat exchange.

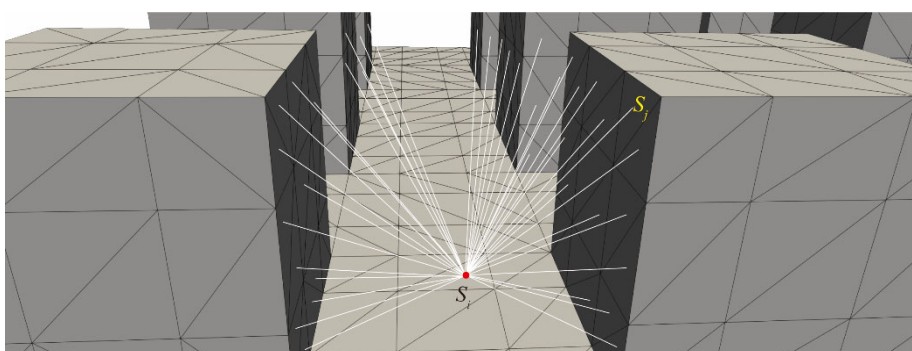

**Figure 6: Schematic illustration of how view factors are calculated between urban surface elements.**

## 2.4. Outdoor convective sub-model

GUST does not calculate urban airflow; instead, it uses empirical formulas to calculate the outdoor convective heat flux as follows:

$$q_{c,out} = U_f h_{out}(T_{w,out} - T_{a,out}) \tag{13}$$

where $T_{a,out}$ is the outdoor air temperature in the canopy layer, $U_f$ is the wind speed, and convective heat transfer coefficient $h_{out} = 4.5 \left(\frac{Ws}{m^3 K}\right)$ is adopted.

The wind speed above the urban canopy layer (UCL) is calculated by a logarithm wind profile:

$$U(z) = \frac{u_*}{\kappa} \ln\left(\frac{z + z_0}{z_0}\right) \tag{14}$$

where $z_0 = 0.1H$ based on the estimation of (Grimmond and Oke, 1999).

The wind speed within the UCL is assumed to be uniform and is calculated by the model by Bentham and Britter (Bentham and Britter, 2003). This model estimates the in-canopy velocity ($U_c$) based on the frontal area density ($\lambda_f$) as follows:

$$\frac{U_c}{u_*} = \left(\frac{2}{\lambda_f}\right)^{0.5} \tag{16}$$

Here, the friction velocity ($u_*$) depends on the urban morphology and is estimated using the following functions (Yuan et al., 2019):

$$\begin{cases} u_* = 0.12U_{2H}, & \text{for } (\lambda_f > 0.4) \\ u_* = 6.7U_{2H}^3 - 6.4U_{2H}^2 + 1.7U_{2H} + 0.03, & \text{for } (\lambda_f < 0.4) \end{cases} \tag{17}$$

where $U_{2H}$ is the wind speed at a height of $2H$ above the ground, and $H$ is the building height.

The air temperature in UCL is assumed to be uniform and calculated by the urban canopy model (Yuan et al., 2020). This model estimates the in-canopy temperature based on the exchange velocity $U_E$ and sensible heat flux $q_{c,out}$.

$$T_c = \frac{1}{D_c} \frac{q_{c,out}}{U_{2H}(1 - \lambda_p)} \left(1 - 0.12\left(\frac{2}{\lambda_f}\right)^{0.5}\right) + T_{a,2H} \tag{18}$$

where $D_c$ = 17.183, is a heat capacity constant of the air, $T_{a,2H}$ is the air temperature above the roof
level, $\lambda_p$ is the plan area density. Bentham and Britter (Bentham and Britter, 2003) suggested that the
$U_E$ can be calculated by:
$$\frac{U_E}{u_*} = \left(\frac{U_{2H} - U_c}{u_*}\right)^{-1} \tag{19}$$

The $q_{c,out}$ is calculated by the temperature from previous time step.

## 2.5. Indoor sub-model

The indoor side uses a convective boundary condition given by $q_{in} = h_{in}(T_{w,in} - T_{a,in})$, where $T_{a,in}$ is
the indoor air temperature, $T_{w,in}$ is the wall temperature on indoor side. The indoor heat transfer
coefficient $h_{in} = 13.5\,\frac{\text{W}}{\text{m}^2\text{K}}$ accounts for both natural convection and longwave radiative heat flux.
For air-conditioned rooms, the indoor air temperature is assumed to be constant at $T_{a,in}$ = 26 °C. In
contrast, for naturally ventilated rooms, the indoor air temperature is assumed to be equal to the in-canopy
air temperature, represented as $T_{a,in} = T_c$.

## 3. Model validation and assessment

### 3.1. SOMUCH measurement

The model is validated by cross-compare with the SOMUCH measurement, which is a scale outdoor
field measurement conducted in Guangzhou, P.R. China (23°1′ N, 113°25′ E) (Hang and Chen, 2022;
Hang et al., 2025; Wu et al., 2024). This measurement provides a quality database for evaluating urban
climate models    (Hang et al., 2024; Chen et al., 2025). The campaign conducted from 29th Jan to 1st
Feb 2021 is used. In that campaign, both surface and air temperatures were measured at high resolution,
making it an ideal database for validating current models.
The geometry of the building blocks and measurement points are plotted in Fig. 7. In that measurement,
the urban buildings are modeled by hollow concrete blocks with a size of 0.5 m× 0.5 m× 1.2 m and a
thick of 0.015 m. The blocks are arranged to form street canyons with four different aspect ratios, i.e.,
H/W = 1, 2, 3, 6. Each row consists of 24 blocks and has a length of L = 12 m. In the experiment, the
surface and air temperatures are measured using thermocouples (Omega, TT-K-36-SLE, Φ0.127 mm and
TT-K- 30-SLE, Φ0.255 mm). The wind speeds inside and above the street canyon are measured using
sonic anemometers (Gill WindMaster). The incoming longwave and solar radiation are measured using
weather stations (RainWise PortLog). The thermal characteristics of the concrete and ground are listed
in Table 1.

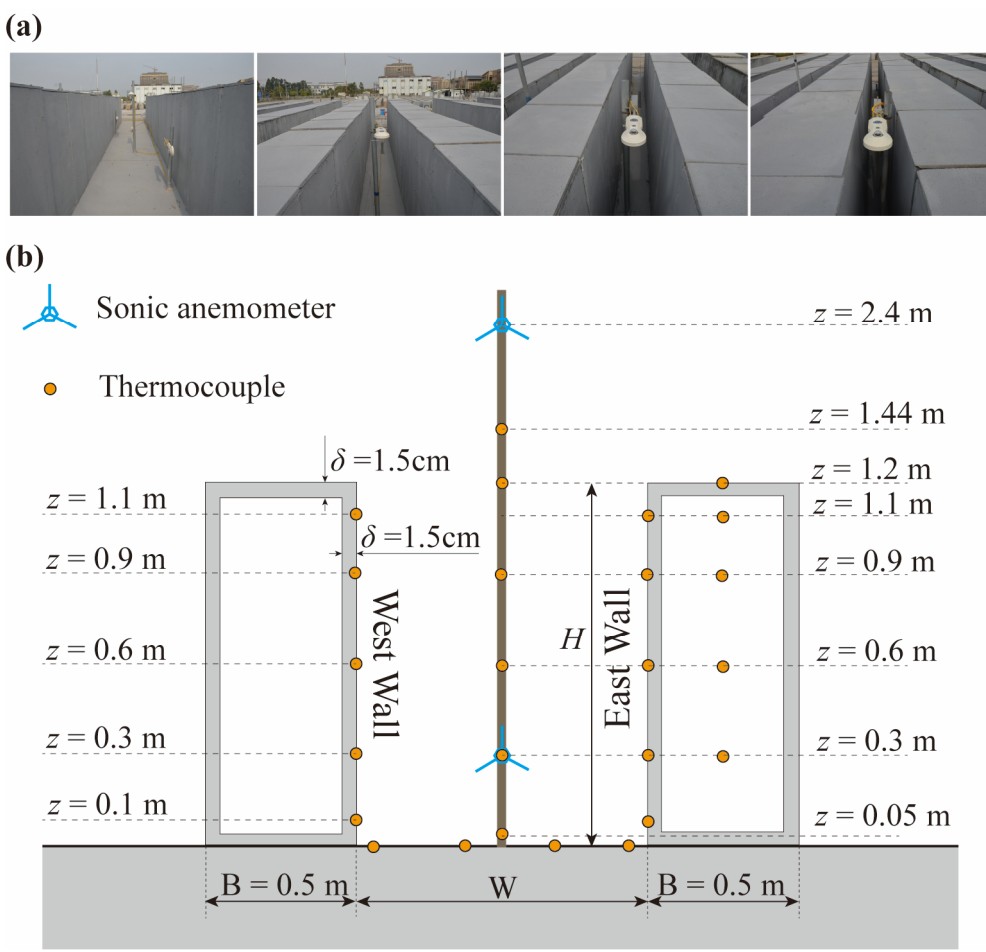


**Figure 7: Photograph of the SOMUCH experiment (a). The geometry of concrete blocks and measurement**
**points in SOMUCH (b). The thermocouples are used to measure the surface temperature and air temperature.**
**The sonic anemometers are used to measure wind speed.**

**Table 2. Thermal properties of the building material. The emissivity is for the longwave radiation and albedo is for the solar radiation.**

| Material | Density $\rho$ | Conductivity $k$ | Specific Heat Capacity $c_p$ | Emissivity $\varepsilon$ | Albedo $\alpha$ |
|---|---|---|---|---|---|
| | (kg m$^{-3}$) | (W m$^{-1}$ K$^{-1}$) | (J kg$^{-1}$ K$^{-1}$) | | |
| Concrete | 2420 | 2.073 | 618 | 0.87 | 0.24 |

## 3.2. Cross comparison of the roof temperature

The surface temperature model is validated by cross-comparing with SOMUCH measurement. Many factors affect the accuracy of the model, including the radiation, convective and conduction. To separately investigate these factors, the temperatures at roofs are first validated because the total radiative flux of roof is only influenced by the incoming longwave and solar radiation. The shading effect of other blocks can be ignored as the block heights are uniform. Therefore, the accuracy of conductive and convective sub-models can be separately evaluated.

The accuracy of this model is quantitatively evaluated by two statistical parameters, the root mean square error (RMSE), and coefficient of determination ($R^2$). The RMSE and $R^2$ of $u_x^*$ are calculated by:

$$\text{RMSE} = \sqrt{\frac{1}{n}\sum_{i=1}^{n}(O_i - P_i)^2} \tag{21}$$

$$R^2 = 1 - \frac{\sum_{i=1}^{n}(O_i - P_i)^2}{\sum_{i=1}^{n}\left(O_i - \overline{O_i}\right)^2} \tag{22}$$

where $O_i$ represents the measured values, $P_i$ is the simulated values, $\overline{O_i}$ is the mean of the measured values, and $n$ is the number of data points.

The wind speed at roof level is needed to calculate the outdoor convective flux of roofs. In SOMUCH measurement, the wind speed was measured above the roof and at a height of $2H$. The wind speed at roof level is estimated by a logarithm wind profile as:

$$U(z) = \frac{u_*}{\kappa} \ln\left(\frac{z + z_0}{z_0}\right) \tag{23}$$

where $z_0 = 0.1H$ based on the estimation of (Grimmond and Oke, 1999). The wind velocity at roof level ($z = H$) can be calculated by $\frac{U_H}{U_{2H}} = 0.787$. The outdoor air temperature, incoming solar and longwave radiation, are from the weather station ($z = 2H$).

For the indoor side, the radiative flux between indoor surfaces is ignored in this model. Only the convective flux is modeled. The convective velocity is assumed to be 3 m/s and CHTC is assumed to be 4.5 for indoor side. Data from the indoor measurement point at $H = 1.1$ m is used. That point is the nearest measurement point to the roof.

Figure 8(a) shows the measurement data that was used to drive the model. During the measurements, the building model was enclosed, leading to the development of very high indoor temperatures. Therefore, the measured indoor air temperature was used as an input for the validation simulation. Fig. 8(b) shows the roof surface temperatures from measurement and simulation. Generally, the roof surface temperatures are well reproduced by the model, because the $R^2$ is 0.99 and $RMSE$ is 1.28. The large discrepancy is found around noon. The model slightly overestimates the roof temperature. The comparison of roof temperatures shows that the conductive and convective sub-models are reliable.

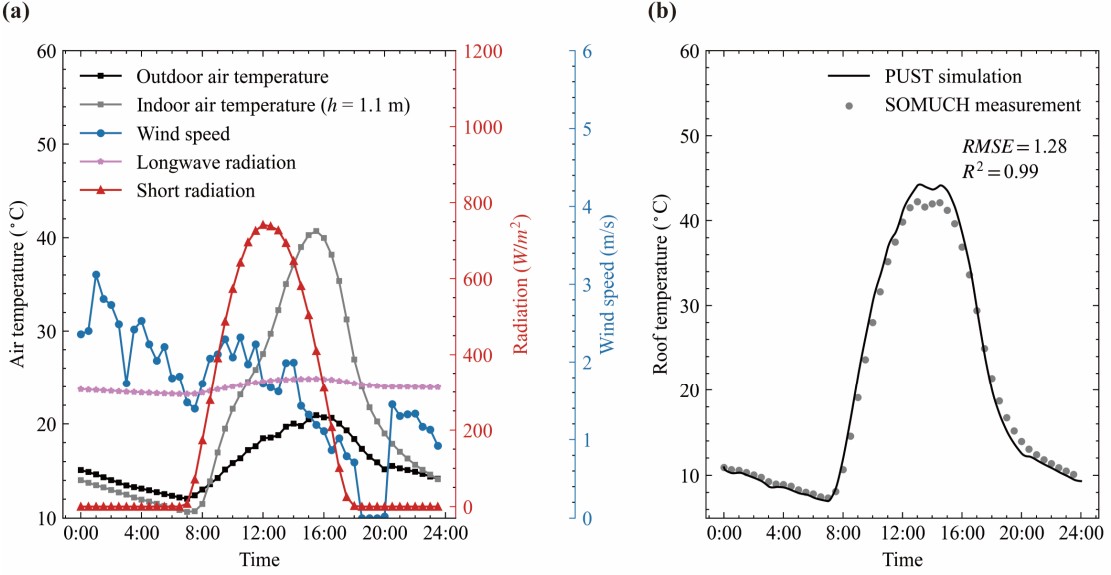

**Figure 8: Weather data on the measurement date (29 January 2021) is shown in (a). Panel (b) compares roof surface temperatures from simulation and measurement, where points denote measured data and lines denote**

**simulated data.**

## 3.3. Cross comparison of the wall temperature

The temperatures at walls are more complicated than those at the roof because the buildings change the
radiative fluxes and wind speeds in street canyons. The radiative fluxes need to be accurately modelled
as they are the main energy input and have a large impact on the surface temperature. To avoid the
influence of air temperature and wind speed modeling, the canyon air temperature, wind speed, and
indoor temperature are from the measurement. The air temperatures are measured from multiple heights.
For the convective flux modelling, the nearest measured air temperatures are used. The wind speeds from
the sonic anemometer in the street canyon ($z = 0.3$ m) are used to calculate the convective flux at outdoor
side. The driving data are plotted in Appendix A.
The east and west walls are defined by taking street canyon center as the origin point. The street direction
is tilted from north toward east by 25°. Therefore, the west and east walls are roughly defined to
distinguish them. The street orientation has been modeled in our model and will not cause additional
discrepancy.
Figures 9 and 10 show the comparison of wall temperatures from simulation and measurement. For each
surface, multiple points are compared to avoid the influence of localized anomalies and to ensure that
the evaluation reflects the overall wall-temperature behavior. Generally, the wall temperatures are well
reproduced, particularly their variation trend. The peak hours are well reproduced. For example, there
are two temperature peaks for the west wall. The first one is around 10:00 and the second is around 16:00.
Both simulation and measurement show the same occurring time.
To quantify model performance, the coefficient of determination ($R^2$) and root-mean-square error
(RMSE) were calculated and marked in each sub-figure. Except for the $H/W$ = 6 case, the $R^2$ values
exceeded 0.9 for all walls, confirming a strong correlation between simulation and measurement. For
$H/W$ = 6, $R^2$ is lower because of nighttime underestimation, although the RMSE remains within the
same range as the other cases (1.6 °C to 2.2 °C). The main reason for this discrepancy is that wall
temperatures in deep street canyons ($H/W$ = 6) show only a slight increase compared to the air
temperature, due to minimal sunlight penetration into the canyon. Under these conditions, wall
temperatures become particularly sensitive to convective and longwave radiative fluxes, which amplifies
the impact of small modeling uncertainties.

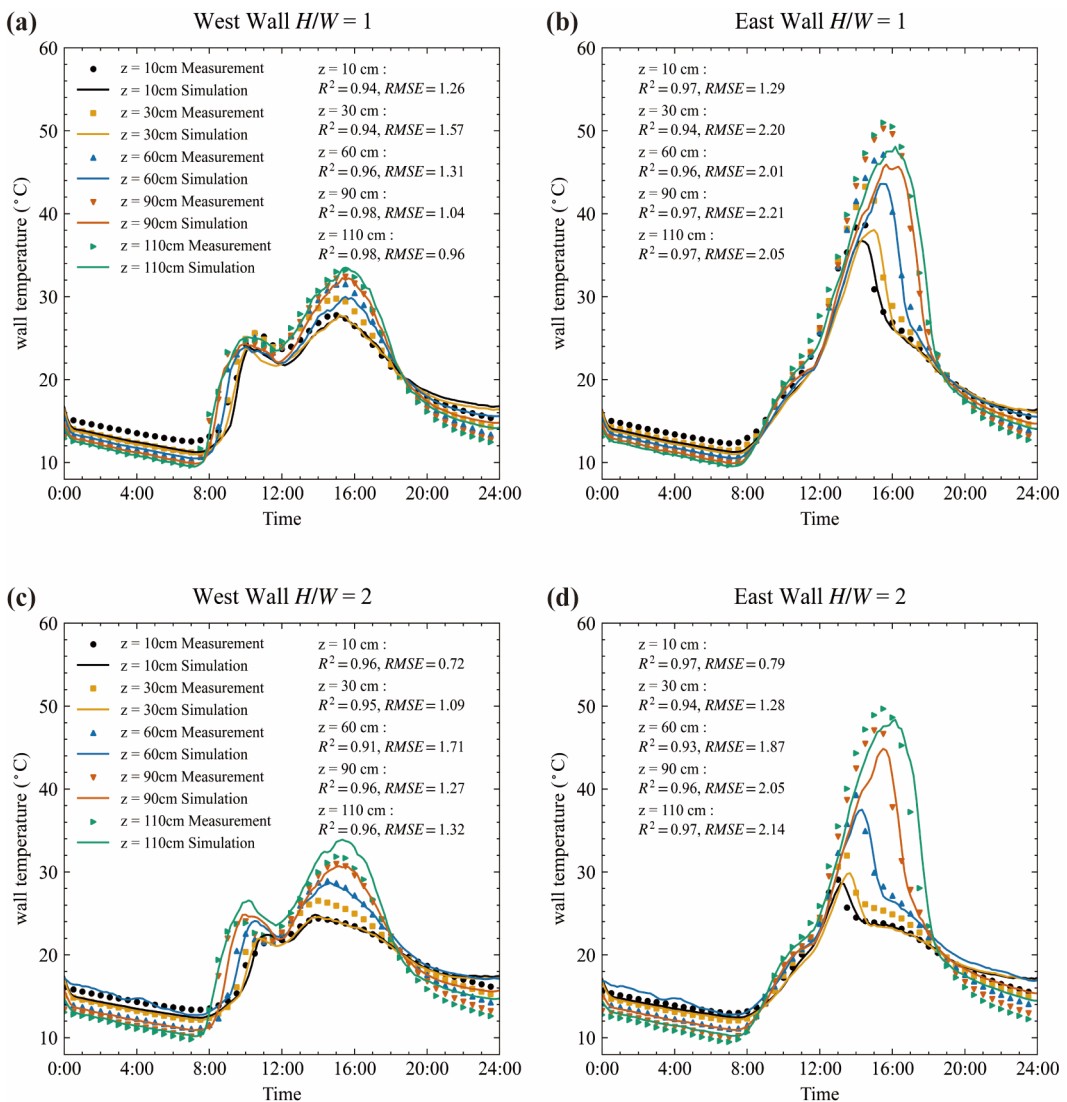


**Figure 9: Wall temperature comparison between simulation and measurements for street canyons with aspect**
**ratios of *H/W* = 1.0 and 2.0. Surface temperatures were measured on 29 January 2021. The root mean square**
**error (RMSE) and coefficient of determination (R²) are calculated and shown. Symbols denote measurements,**
**while lines indicate simulations. The left panel corresponds to west side walls and the right panel to east side**
**walls.**

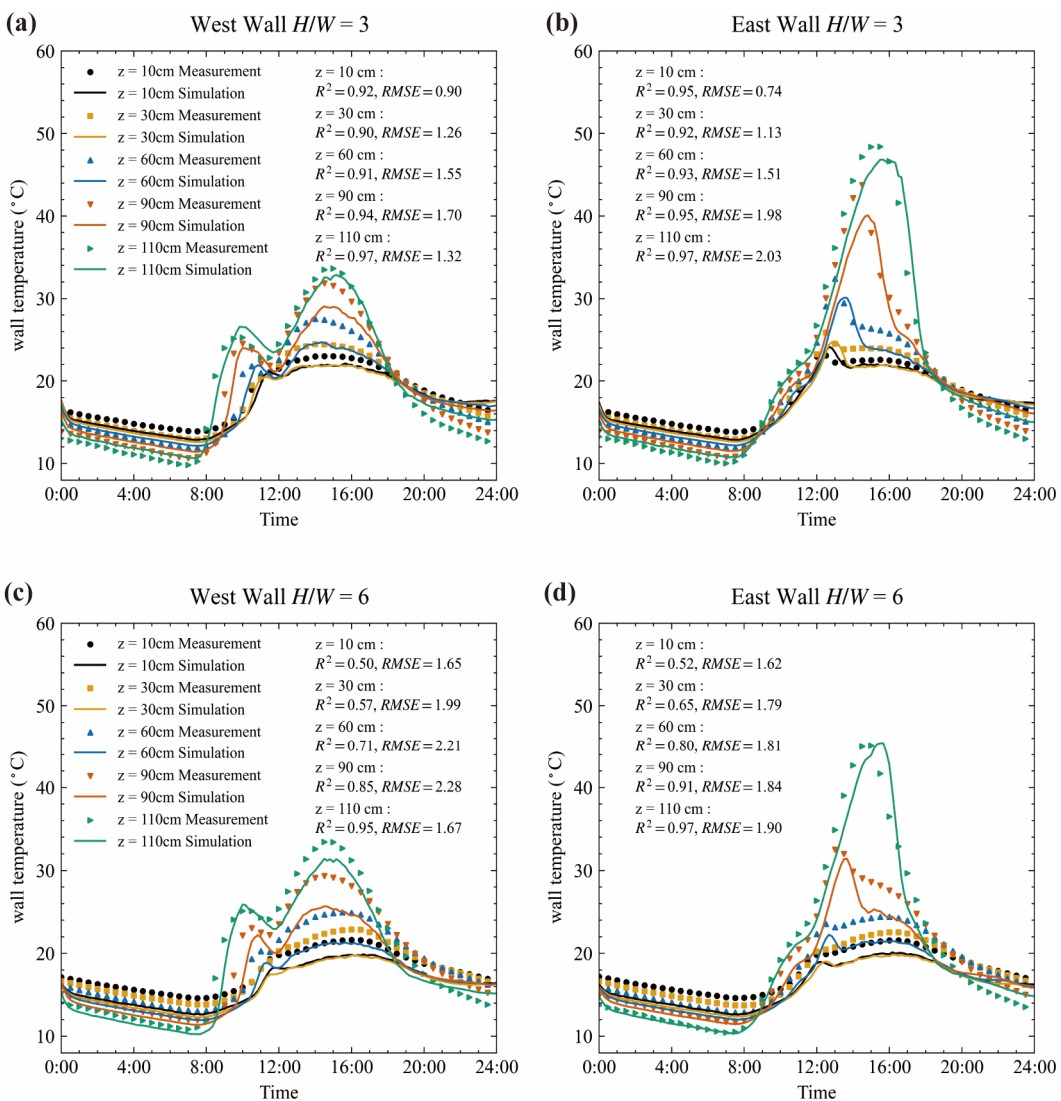

**Figure 10: Wall temperature comparison between simulations and measurements, as in Figure 9, but for street canyons with aspect ratios of *H/W* = 3 and 6.**

## 3.4. Cross comparison of the ground temperature

The surface temperatures of the ground are heavily influenced by heat storage. During the day, heat is conducted to deeper layers and stored there. At night, this stored heat is released. Therefore, the initial temperature field and boundary conditions are critical for accurately modeling surface temperatures. In this study, an adiabatic boundary condition is applied at a depth of 0.5 m below the ground surface. The soil material is divided into three layers with thicknesses of 0.2 m, 0.15 m, and 0.15 m. All three layers are assumed to be made of concrete. The thermal properties in Table 1 are used. The underground

temperatures are measured by thermocouples with three depths of 5 cm, 10 cm, and 20 cm, as plotted in
Appendix A.  In this study, we used only the measured underground temperatures at 0:00 to initialize the
underground temperature field. It is important to note that the available soil temperatures were measured
in open ground rather than under street canyons. This difference may lead to discrepancies in modeling
ground surface temperatures.
Figure 11 shows the ground surface temperatures from measurement and simulation. The ground surface
temperatures are measured at four locations: g1, which is close to west wall; g4, which is close to east
wall; and g2 and g3, which are situated in the middle of the streets. Generally, the temperature variations
are well reproduced by the model. For example, peak temperatures occur sequentially from g1 to g4 due
to the movement of the building's shadow. This phenomenon is observed in both simulations and
measurements.
The accuracy of ground temperatures is lower than that of the wall temperatures in terms of $R^2$. For
example, in $H/W = 2$, the R² values for temperatures at the west wall range from 0.91 to 0.97, while
those at the ground range from 0.67 to 0.90. However, the ground temperatures can be considered well
modeled because the RMSE for ground temperatures is smaller than that for wall temperatures. Using
$H/W = 2$ as an example, the RMSE values for the west wall range from 0.69 to 1.71 °C, while those for
the ground range from 0.98 to 1.37 °C. This difference between the R² and RMSE values is due to the
ground temperature increase being much lower than that of the walls because of shading, particularly in
deep street canyons.
Uncertainties in the input data may also contribute to the discrepancies between simulation and
measurement. First, the thermal properties of soil can differ significantly from those of concrete blocks.
Secondly, the initial temperature is measured in surrounding area, rather than in street canyons. Thirdly,
since the same initial temperature field is used for all four points, the model is unable to reproduce the
differences between points at night.

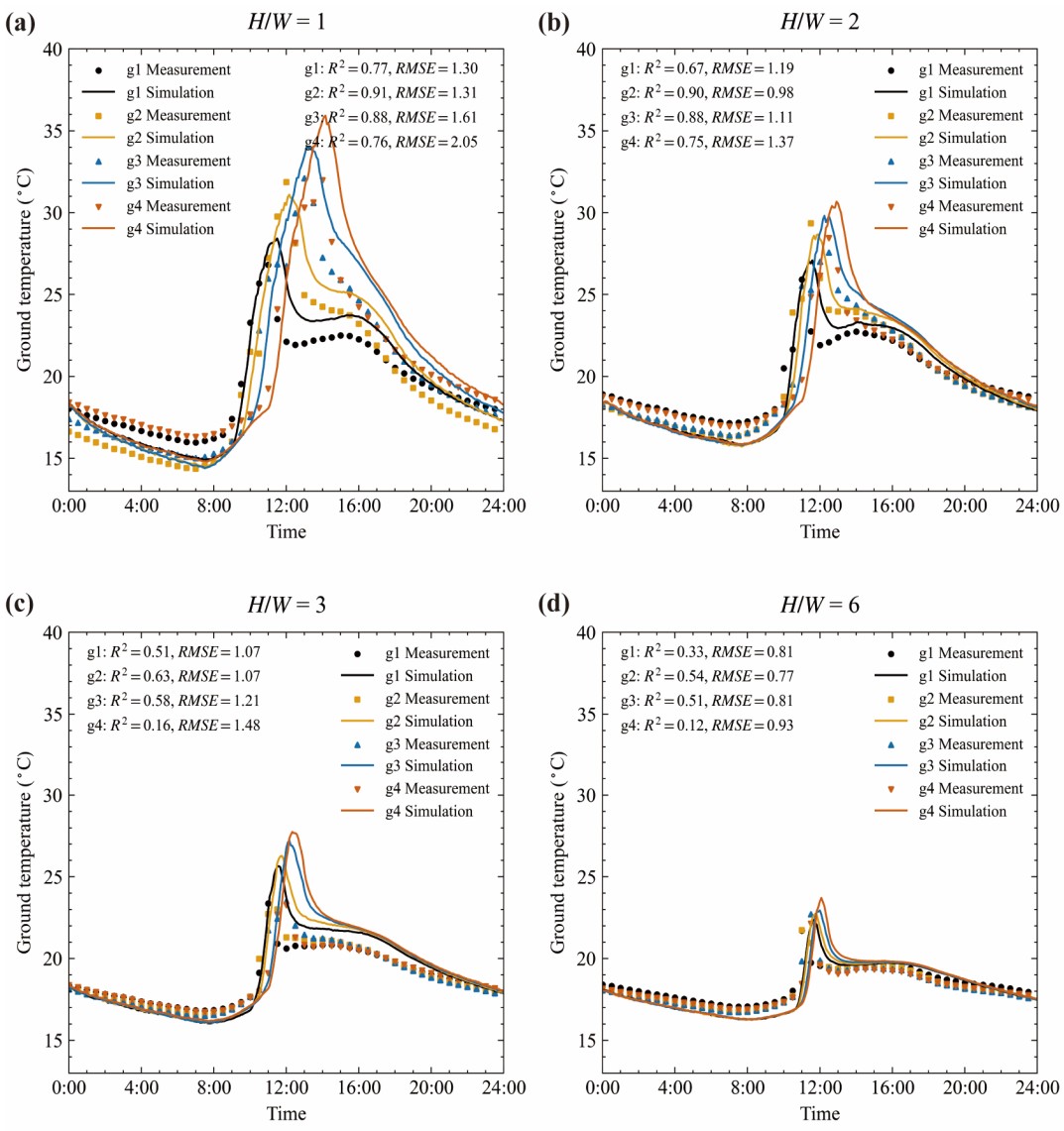


**Figure 11: Ground temperature comparison between the simulation and measurement results at street canyon aspect ratio of *H/W* = 1.0, 2.0, 3.0, and 6.0. Surface temperatures are measured on 29th Jan 2021. The root mean square error (RMSE), and coefficient of determination (*R²*) are calculated and plotted. The points represent measured data and lines represent the simulated data.**


## 3.5. Surface energy balance analysis

The surface temperature comparison indicates that model uncertainties arise from various factors. To identify the main factors impacting the model accuracy, the energy balance of wall surface is analyzed.

The heat fluxes of solar ($Q_K$), longwave radiation ($Q_L$), convection ($Q_H$), and conduction ($Q_G$) of outer
surface of walls satisfy the following equation:
$$Q_K \ + \ Q_L \ + \ Q_G \ + \ Q_H \ = \ 0 \tag{24}$$
Here, the longwave heat flux $Q_L$ is divided into two parts as the heat exchange between wall to sky
($Q_{L,sky}$) and to other urban surfaces ($Q_{L,urban}$), expressed as $Q_L = \ Q_{L,sky} + Q_{L,urban}$. This analysis aims
to determine whether it is necessary to model the longwave heat exchange between urban surfaces, which
requires substantial computational resources.
Figures 12 and 13 show the heat fluxes of walls in the simulation. The heat fluxes of east and west walls
are averaged from five measurement points on each. Our previous work (Mei et al., 2025) demonstrated
that the MCRT can accurately predict solar radiation in high-density urban configurations, while also
achieving high computational efficiency through GPU-based acceleration. In that study, we compared
the albedo of the urban canopy layer and of street canyons across a range of urban layouts with in-situ
measurements, achieving excellent agreement. The previous study also serves as an independent
validation of the ray-tracing component within the modeling framework. Although the ray-tracing
procedure in the present study differs from that in our previous work, the core computational framework
remains the same. In the previous study, solar rays were emitted directly from the sun and sky, whereas
in this study, we adopted a reverse ray-tracing technique, in which rays are emitted from building surfaces
toward the surrounding environment.
In all cases, longwave radiative heat exchange between urban surfaces plays an important role in the
energy balance, particularly at high aspect ratios. The longwave radiative fluxes from sky only contribute
a small amount of total longwave radiative flux in $H/W = 6$, as shown in Fig. 12(d) and Fig. 13(d). The
shading effect of buildings creates heterogeneous surface temperatures within the urban canopy layer.
The large temperature differences between surface elements contribute a large portion of the total heat
flux. This highlights the necessity for accurate modeling of longwave heat exchange between urban
surfaces, even though it demands significant computational resources.
The conductive heat flux also contributes a large portion of the total heat flux. It is negative in the
morning and positive in the afternoon, meaning that heat is stored in the building block during the
morning and released in the afternoon. In the reduced scale experiment, buildings were represented by
airtight hollow concrete blocks. Due to the lack of ventilation, the indoor air temperature can rise to 40°
C under an outdoor air temperature of 20°C, as shown in Appendix A. This indicates that the indoor air
can also absorb, store, and release a considerable amount of heat. Therefore, accurately modeling indoor
air temperature is essential for effective surface temperature modeling.
The convective heat flux contributes a smaller amount of the total heat flux. In high aspect ratio cases
($H/W$ = 3 and 6), the convective heat fluxes are almost negligible. This is due to the weak wind in the
deep street canyons. In this model, the surface convective heat flux is directly calculated from the wind
speeds in street canyons. This assumption may underestimate the convective flux, especially since natural
convection occurs under weak wind conditions (Fan et al., 2021).

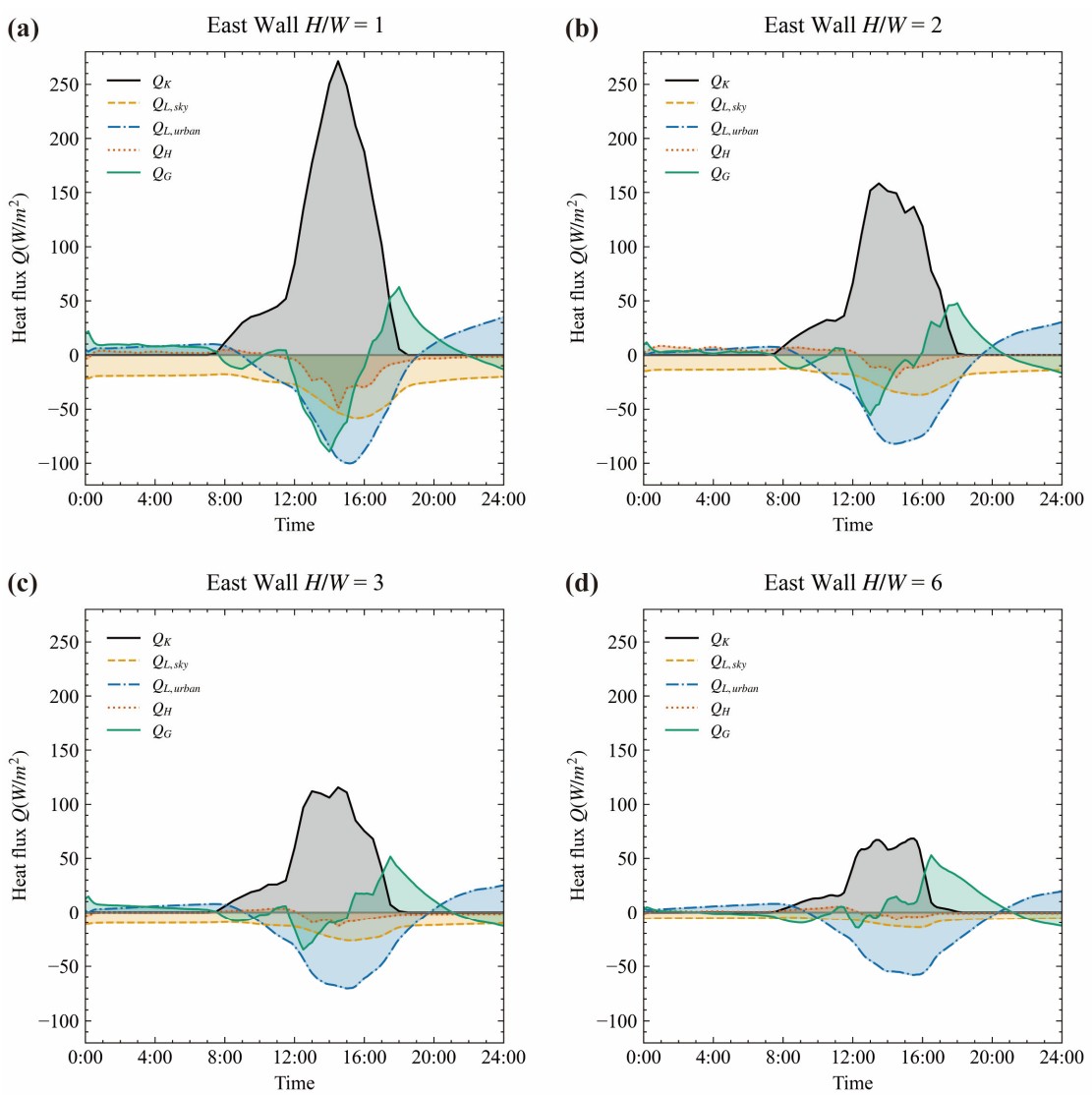


**Figure 12: Diurnal heat fluxes at the east side walls from the simulation. The heat fluxes of solar ($Q_K$),**
**longwave radiation ($Q_L$), convection ($Q_H$), and conduction ($Q_G$) are at the outer surface of walls.**

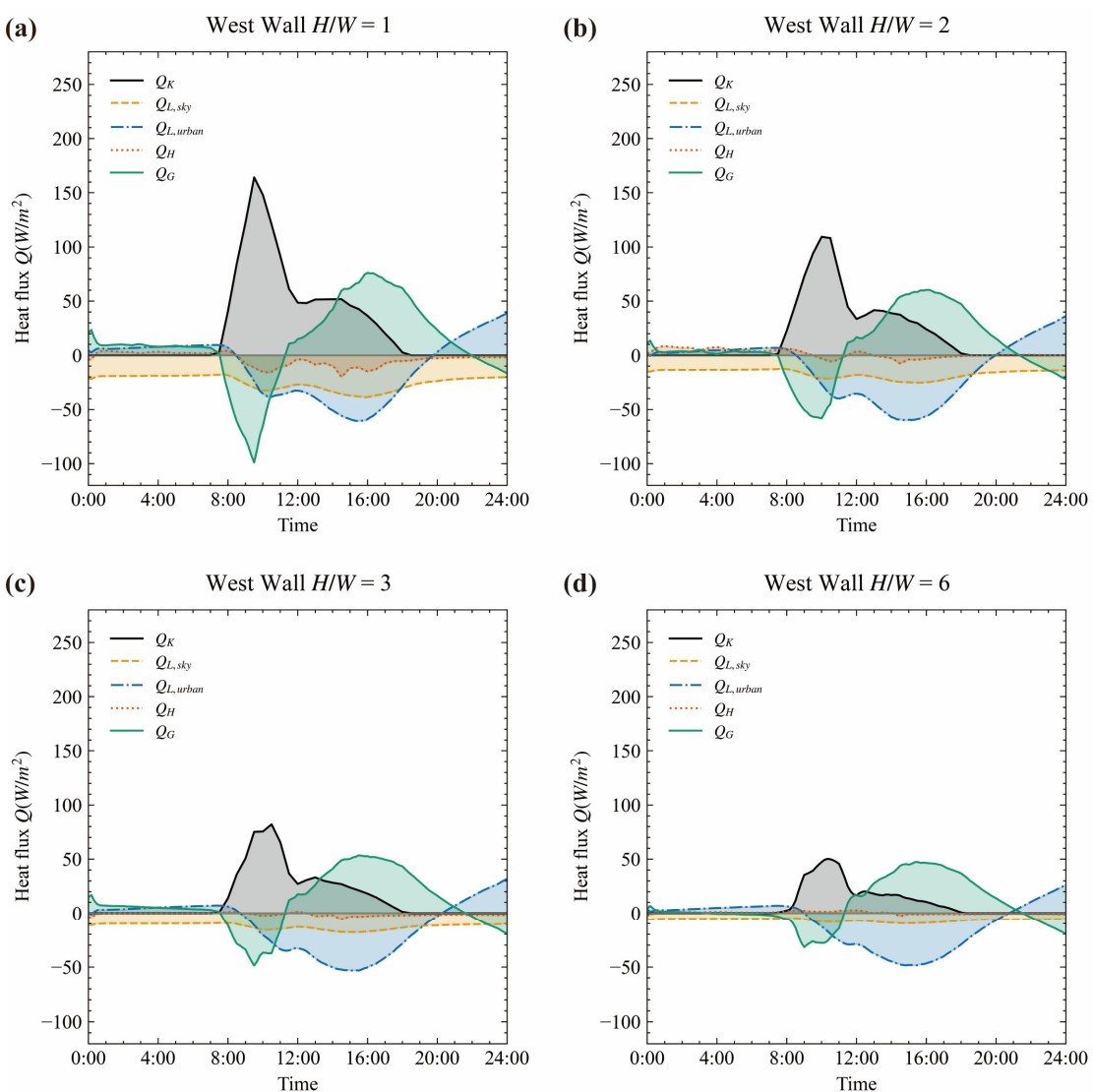


**Figure 13: Diurnal heat fluxes at the west side walls from the simulation. The heat fluxes of solar ($Q_K$),**
**longwave radiation ($Q_L$), convection ($Q_H$), and conduction ($Q_G$) are at the outer surface of walls.**
**4. Application to real urban configuration**
To demonstrate the model's applicability to complex geometries, we simulated a neighborhood
containing 40 buildings within an area of 350 m × 200 m. Building geometries were imported as STL
files comprising approximately $2.3 \times 10^4$ triangular surface meshes. Surface temperatures were calculated
on the triangular surface elements, as shown in Fig. 6, with shortwave fluxes resolved by a Monte Carlo
ray-tracing scheme using $1 \times 10^5$ photons. The solar position is updated at 30-min intervals to capture both
diurnal and shading variations. Transient heat conduction simulations were performed for 24 h with a
10-min time step (600 s) on 29 January 2021, consistent with the validation case. Downward solar
radiation, longwave radiation, wind speed, and air temperature were prescribed from the SOMUCH
measurements.
The simulation ran on a local workstation with an NVIDIA RTX 5090D GPU and completed in 26.6 h,
comprising a view-factor calculation (4.2 h), solar-radiation computation (22.2 h), and coupled heat-
transfer analysis (0.2 h).
For this demonstration, material-specific reflectance was neglected and a uniform albedo of 0.24 was
applied to all urban surfaces. Walls and roofs were modeled as three concrete layers of 0.10 m each (total
thickness = 0.30 m), while the ground comprised 0.35 m (0.15 m + 0.15 m + 0.05 m) with an adiabatic
bottom boundary. For all layers, thermal properties were fixed to concrete values of thermal conductivity
$k = 2.0\ \mathrm{W\,m^{-1}K^{-1}}$, density $\rho = 2420\ \mathrm{kg\,m^{-3}}$, and specific heat capacity $c_p = 618\ J\,\mathrm{kg^{-1}K^{-1}}$. All
model inputs are consolidated into a single YAML configuration file, which specifies the simulation
parameters, weather forcing, geometry paths, surface albedo, and material thermal properties for easy
reproducibility. The buildings are assumed to be naturally ventilated, with the indoor and outdoor air
temperatures being the same. The thermal characteristics of concrete are assumed to be the same as in
the SOMUCH experiment.
The surface temperatures are calculated in three steps: 1) calculate the solar radiative flux of each point
by rMCRT; 2) calculate the view factors between the elements using rMCRT; 3) calculate the surface
temperatures using Monte Carlo random walking. All three steps are processed in parallel on GPU. The
weather data measured on 29th Jan 2021 during the SOMUCH experiment is used as the driving input.
The surface temperatures are calculated from 0:00 to 24:00, with a time step of 30 minutes.
The simulation results were exported in vtk format and visualized using ParaView. Fig. 14 presents the
surface temperature distributions at 09:00, 11:00, 13:00, 15:00, 17:00, and 19:00. The movement of
building shadows and their influence on surface temperatures are clearly visible in these contours,
illustrating the diurnal heating and cooling cycle. These visualizations demonstrate that the model can
represent complex building geometries and can be applied to real urban environments.
The energy balance analysis of the SOMUCH experiment indicates that convective heat transfer plays
only a minor role. However, due to the experiment's reduced scale and limited local wind speeds, it
remains uncertain whether this conclusion holds at full scale or under higher wind speed conditions.

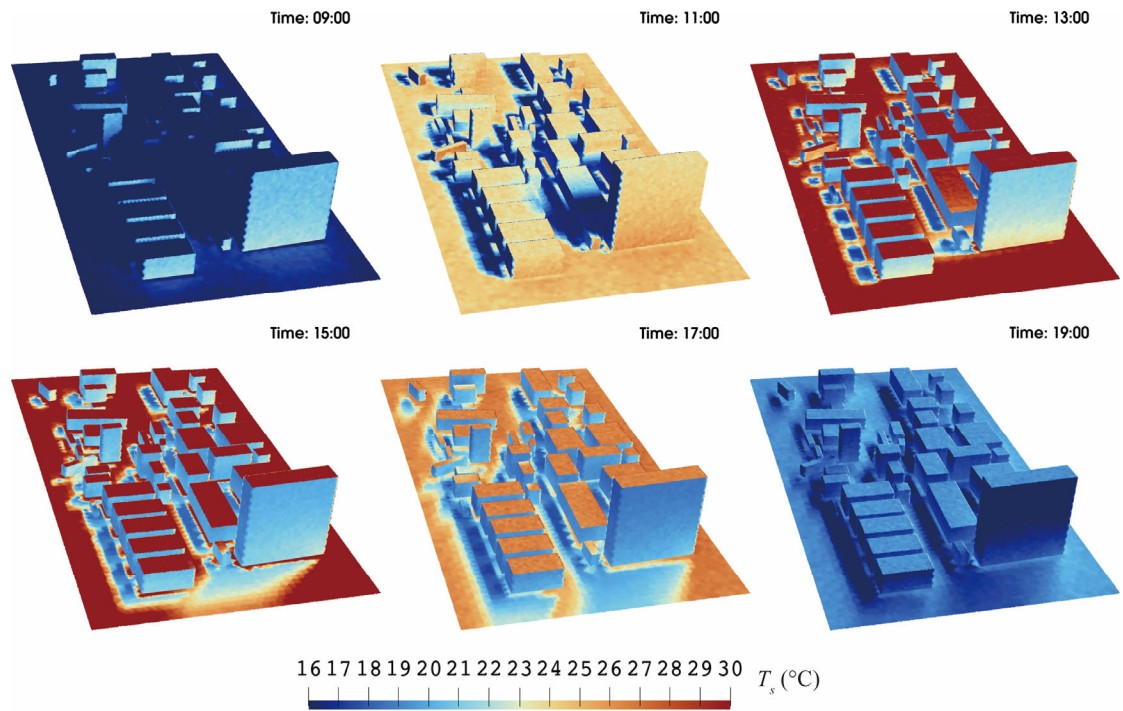


**Figure 14: Simulation results show the evolution of surface temperature for the complex building geometries**
**at 09:00, 11:00, 13:00, 15:00, 17:00, and 19:00. These snapshots capture the diurnal heating and cooling cycle,**
**highlighting morning warming, peak midday temperatures, and the evening decline.**
To further assess the role of the convective model, a wind sensitivity analysis was performed for the real
urban configuration. The baseline wind speed (WF = 1.0) was measured on 29 January 2021, the same
day used for the validation cases. Wind speeds were then systematically increased by factors of 2.0 and
5.0 relative to the baseline to evaluate their influence on urban surface temperatures. The resulting
average surface temperatures of the ground, walls, and roof are shown on Fig. 15. The temperature
evolution in Fig. 15 (a)–(c) demonstrates that increasing the wind factor from WF = 1.0 to 5.0
progressively lowers surface temperatures across all urban elements. Fig. 15 (d) quantifies the
temperature differences relative to the baseline scenario (WF = 1.0), revealing cooling effects of up to
6 °C, with the most pronounced reductions occurring during peak heating hours. Among the three
surfaces, the roof exhibits the greatest sensitivity to wind variations, followed by the ground and then the
walls.
These results highlight that, at full scale and under high-wind conditions, convective processes can exert
a much stronger influence on urban surface temperatures than indicated by the scaled SOMUCH
experiment. Therefore, future studies are needed to better quantify and model convective effects across
a broader range of wind speeds and length scales. Moreover, under weak-wind conditions, natural
convection becomes especially important, particularly when the temperature difference between the wall
and the atmosphere grows large (Fan et al., 2021; Mei and Yuan, 2021). However, this natural-convective
effect may not be significant in the scaled SOMUCH experiment.

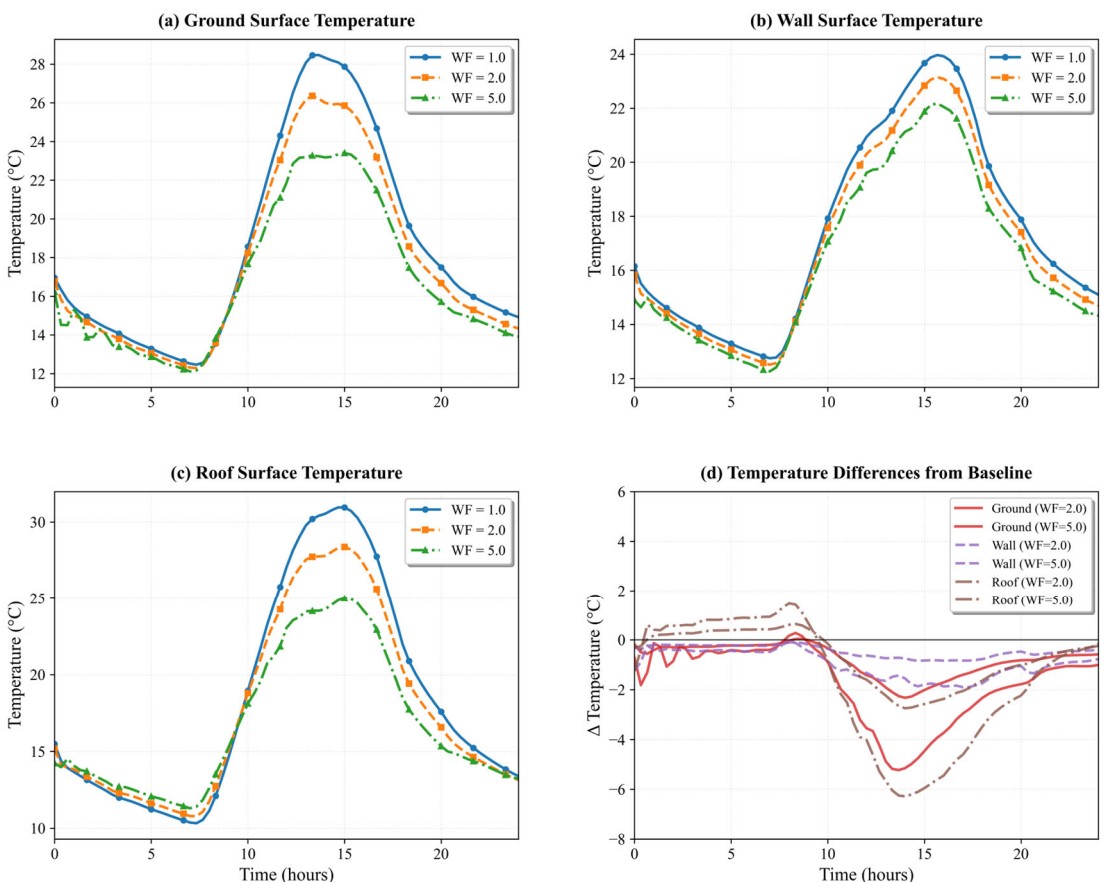


**Figure 15. Wind-sensitivity analysis of urban surface temperatures showing (a) ground, (b) wall, and (c) roof**
**temperature evolution under different wind factors (WF = 1.0, 2.0, 5.0), and (d) temperature differences**
**relative to the baseline (WF = 1.0). The baseline wind speed was measured on 29 January 2021, the same day**
**used for the model-validation cases.**

## 5. Limitations and future work

This model is a building-resolved urban surface temperature model, focusing on detailed neighborhood-scale processes. Therefore, its application to full city-scale simulations remains limited by computational cost and is currently best suited for neighborhood-scale. The first version focuses on the complex radiative exchange in densely built urban areas. The parameters and assumptions are validated against the idealized scaled outdoor experiment, which uses homogeneous building materials with consistent albedo and thermal characteristics. Glazing and green infrastructure are not included in this experiment. The SOMUCH project is currently measuring the impact of glass and green infrastructure. The next version will expand its capabilities to capture complex urban materials, such as urban trees, green walls, and glass curtain walls, to better represent real urban configurations. Other limitations include:

- All reflections are assumed to be Lambertian. While this assumption works well for the SOMUCH measurements, where concrete is used for all urban surfaces, it may not fully capture the reflective properties of other materials with different surface textures, such as glass or vegetation.

- The high-resolution wall temperature simulation still requires a significant amount of time to complete, even with parallel computation on GPUs. This is due to the large number of rays ($N = 10^6$) required for accurate solar radiation modeling. For each point, the simulation takes about 1 second to finish. However, as the number of test points increases, the overall computational time grows substantially.

- The dynamic indoor air temperature is not included in this model. It assumes that the indoor air temperature is equal to the outdoor air temperature for a natural ventilated room. This assumption may lead to discrepancies, particularly in situations where indoor temperatures differ from outdoor conditions due to factors such as heat sources, insulation, or limited ventilation.

- The participation of the urban atmosphere is ignored in this study. In the scaled measurements, longwave radiation travels much shorter distances to adjacent surfaces, which reduces the influence of atmospheric effects compared to real-world urban environments.

Although many additional features will be incorporated into the GUST model in future developments, this does not imply that the current version lacks applicability to real-world scenarios. First, by focusing

on the coupled radiative–convective–conductive heat transfer processes, GUST effectively identifies the
key physical mechanisms responsible for high urban surface temperatures. Second, it provides high-
quality building surface temperature predictions, which can be directly utilized for building energy
consumption analyses. Third, the inclusion of longwave radiative exchange between urban surfaces
enables GUST to be applied in the parameterization of longwave heat fluxes within mesoscale urban
climate models.

## 593   6. Conclusions

This study introduces a GPU-accelerated Urban Surface Temperature model (GUST), which computes
radiation using Monte Carlo ray tracing and solves heat conduction with a one-dimensional Monte Carlo
random-walk approach. To meet the substantial computational demands of these Monte Carlo
simulations, the model employs GPU-based parallel computing for efficient processing. GUST is
validated against the high-resolution, scaled outdoor experiment SOMUCH, which provides detailed
spatial and temporal measurements.
To accurately reproduce multiple reflections in high-density urban areas, the radiative heat flux is
simulated using a reverse Monte Carlo Ray Tracing method. Sensitivity tests show that $10^5 \sim 10^6$ rays
are required for each point to accurately model the solar radiation. This large computational demand for
ray tracing is addressed using GPU-based parallel computing. In addition, the GPU is utilized to
parallelize both the transient heat conduction, which is solved through random-walk algorithms, and the
longwave radiative exchange, which is also computed via ray tracing. This integrated GPU-accelerated
framework substantially improves the computational efficiency and scalability of the GUST model.
The comparison with the SOMUCH experiment shows that the transient surface temperatures on roofs,
walls and the ground are well reproduced. This comprehensive validation demonstrates the model's
ability to accurately capture the fine-scale radiative–convective–conductive heat transfer processes
within complex urban configurations. By conducting a surface energy balance analysis, this study
demonstrates that longwave radiative exchange between urban surfaces plays a critical role across all
building density levels. In contrast, convective heat flux becomes significant only in high-density
configurations.
Lastly, this model is implemented to solve the surface temperatures on complex urban buildings, which
are composed of a total of $2.3 \times 10^4$ surface elements. The GPU allows simultaneous simulation of
heat transfer and view factors across all elements, enabling high-fidelity simulations in real urban
configurations with complex geometries. The current version focuses on the radiation-conduction-
convection coupled heat transfer coupled in complex geometries. Future developments will prioritize the
integration of complex glazing systems and green infrastructure in urban environments.

## Code availability

The SOMUCH measurement data are available upon request. The development of GUST, model
validation, and visualization in this study were conducted using Python 3.8 with CUDA. The source code,
supporting data, and simulation results presented in this paper are archived on Zenodo (Mei, 2025) at
https://doi.org/10.5281/zenodo.17138571 and are freely accessible for research purposes under the
Creative Commons Attribution 4.0 International (CC BY 4.0) license.

## Author contributions

SM designed the study, developed the code, conducted the analysis. SM and GC prepared the manuscript
draft. GC and JH collected and shared SOMUCH measurements for the purpose of model validation. GC,
JH and TS supported the model implementation and data analysis. All have read and accepted the
manuscript for submission.

## Acknowledgement

This research is supported by National Natural Science Foundation of China (Grant No. 42305076,
W2421048, U2442212), Natural Science Foundation of Guangdong Province, China (Grant No.
2024A1515010173) and Overseas Postdoctoral Talents 2023 Programme (Grant No. BH2023009). Dr.
Shuo-Jun Mei and Dr. Ting Sun are supported by an International Exchanges grant from the Royal
Society (Grant No. IEC\NSFC\242040) and National Natural Science Foundation of China (Grant No.
W2421048).

## Appendix A. Indoor and outdoor air temperatures in SOMUCH measurement

The indoor and outdoor air temperatures at different levels in the SOMUCH measurement are plotted in
Fig. A1. These air temperatures serve as input data for the validation cases.

(a) *H/W* = 1

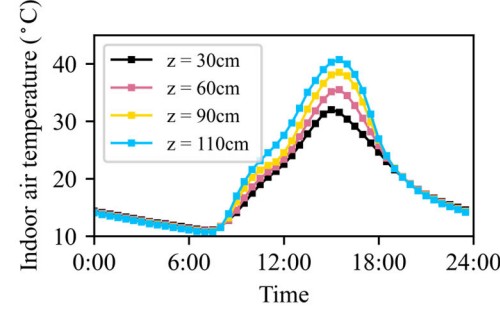
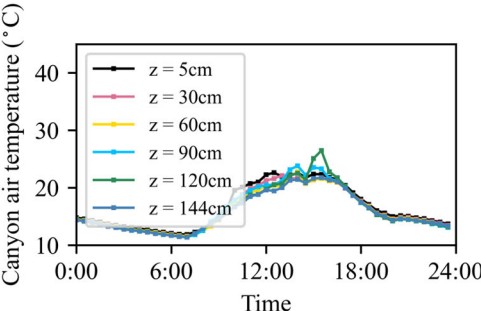
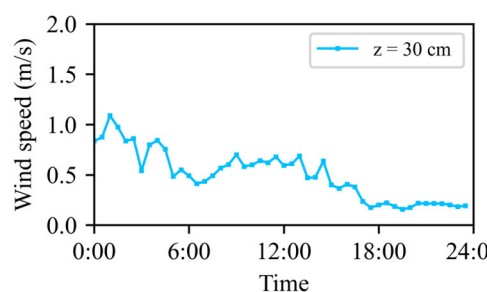
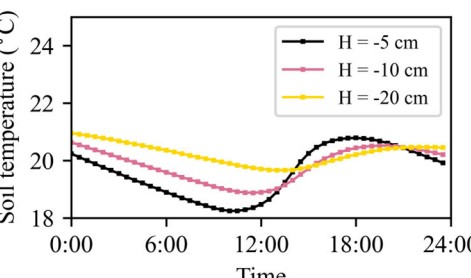


(b) *H/W* = 2

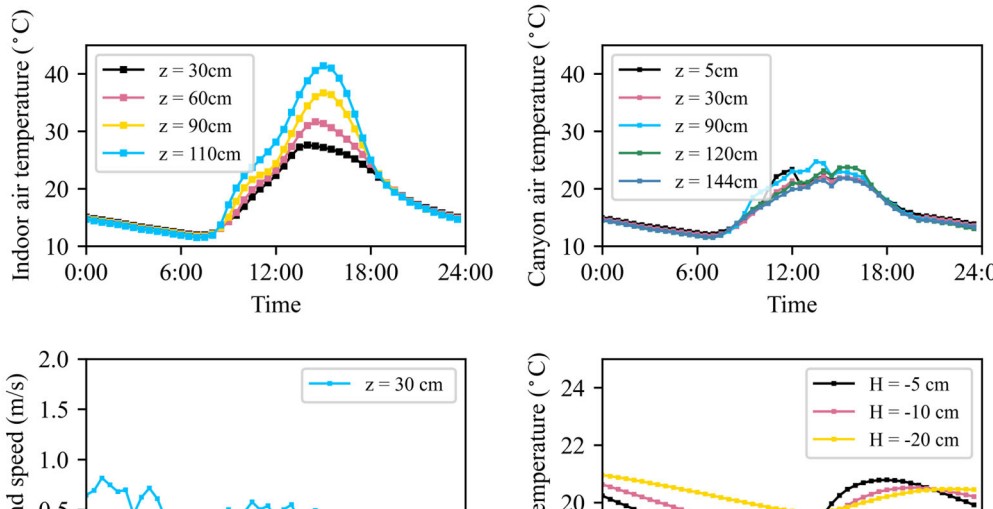


(c) *H/W* = 3

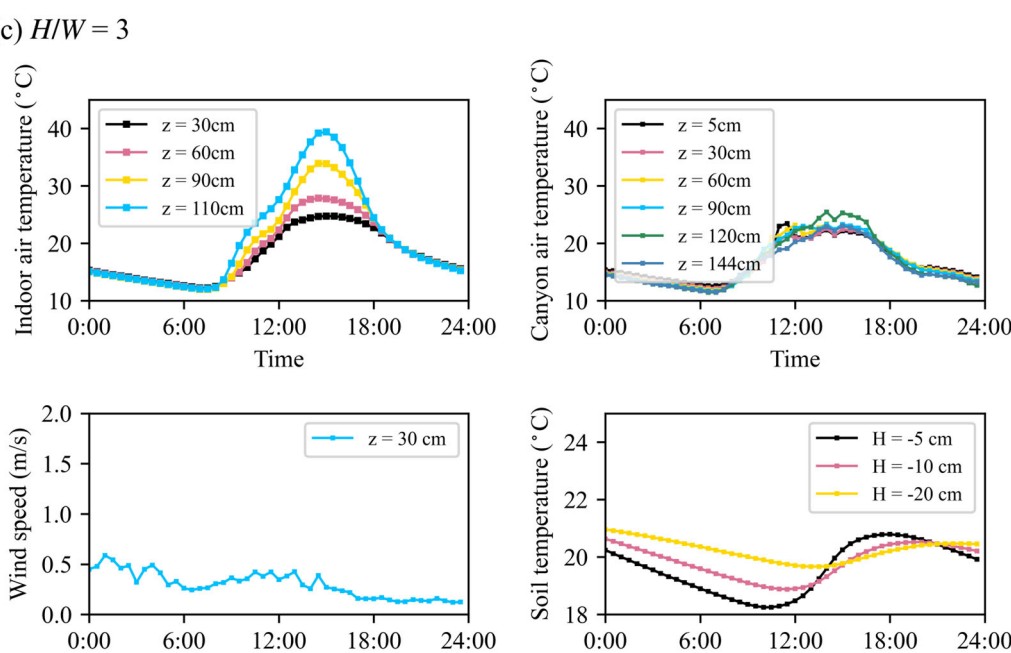


(d) *H/W* = 6

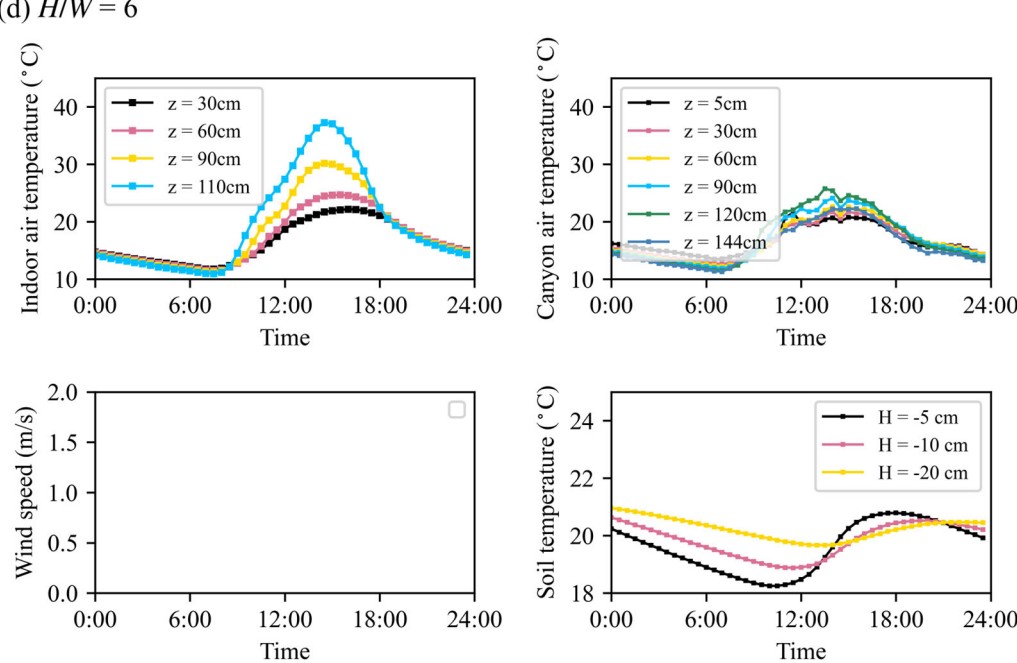


**Figure A1: Indoor, outdoor air temperatures, and wind speeds in street canyons that are measured on 29ᵗʰ Jan 2021. The wind speeds in the street canyon of *H/W* = 6 were not measured because the sonic anemometer cannot be installed in such a narrow street. The outdoor air temperatures measured at z = 60 cm in *H/W* = 2 are unusual, due to an instrument failure.**

## Appendix B. Sensitivity test for other days

To further validate the model, we also compared the simulated roof temperatures with measurements over three consecutive days, from 30 January to 1 February 2021, similar to the analysis presented in Fig. 8. The results are shown in Fig. A2, which demonstrates excellent agreement between simulated and observed roof temperatures. By using multiple consecutive days, this comparison minimizes potential bias arising from the single day's weather conditions.

**(a) 30ᵗʰ Jan 2021**

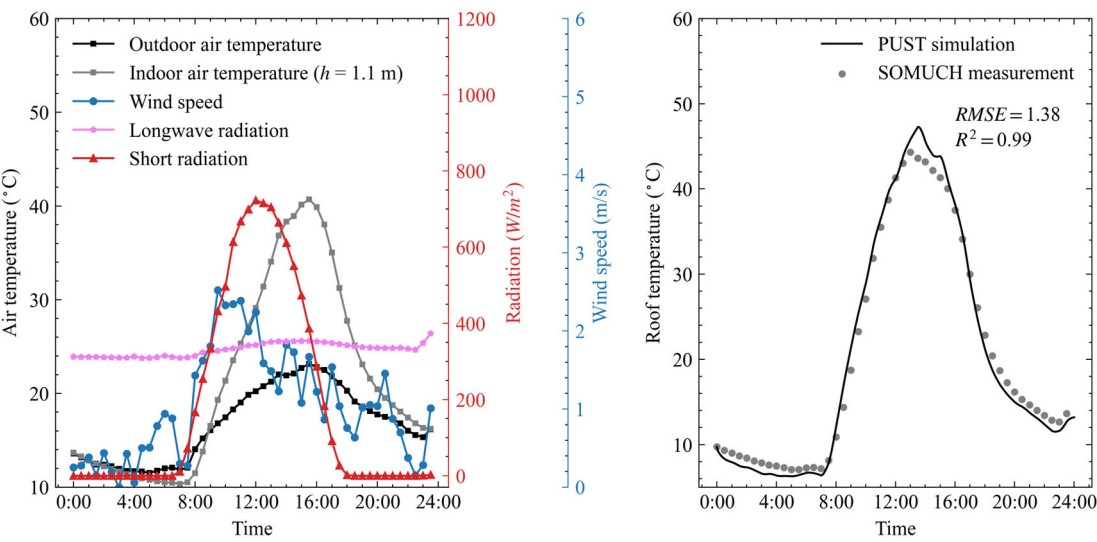

660

**(b) 31ˢᵗ Jan 2021**

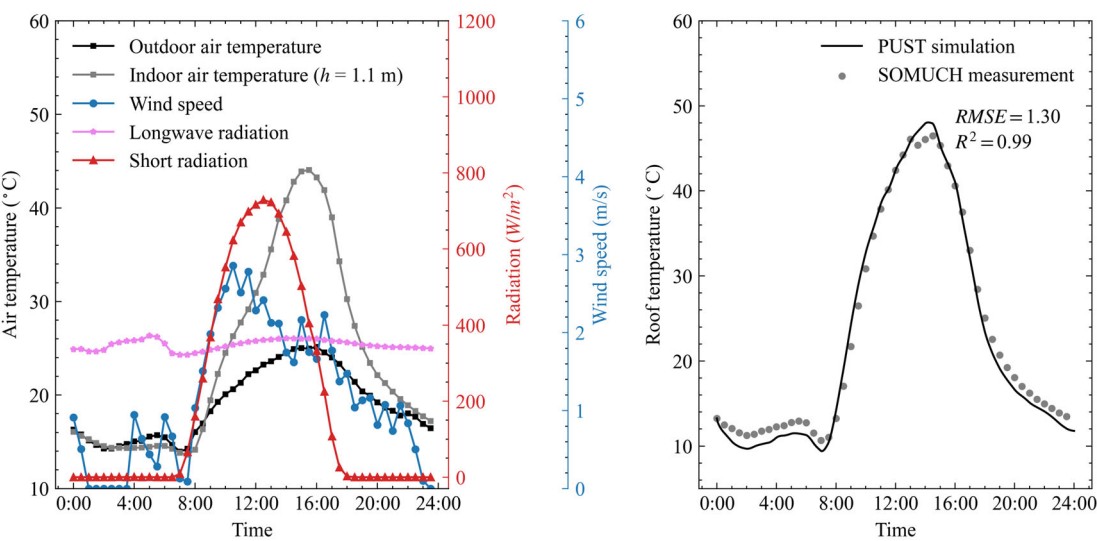

662

**(c) 1ˢᵗ Feb 2021**

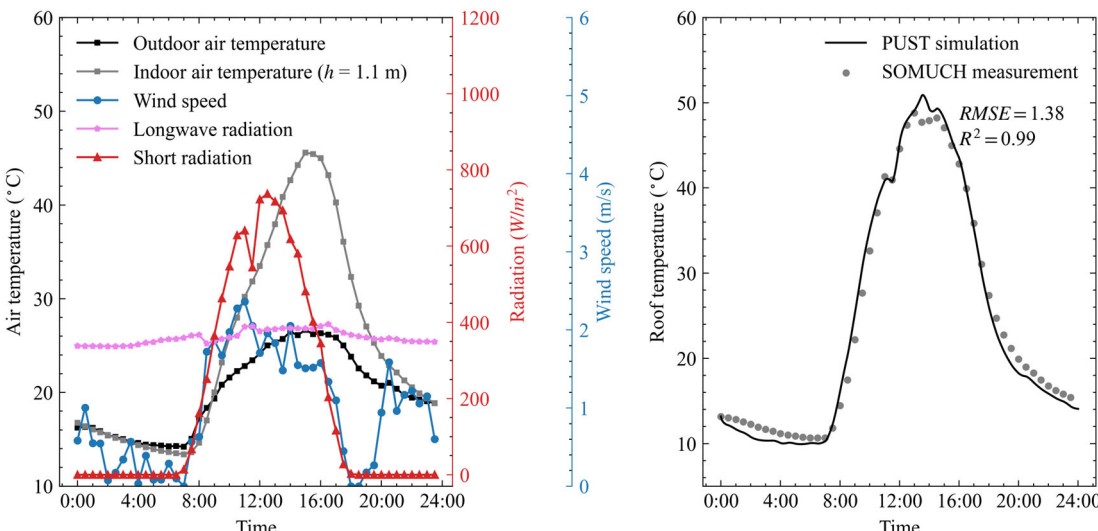

**Figure A2: Weather data from 30 January to 1 February 2021 are shown in the left panels. The right panels compare roof-surface temperatures from simulation and measurement, with points representing observations and lines representing simulated values.**

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
