# Peer review of "GUST1.0: A GPU-accelerated 3D Urban Surface Temperature Model"

_EGUsphere, 2025_

## Author Comment (AC1)

**Reply to RC1**

**General statement**

The manuscript presents GUST1.0, a model for simulating urban surface temperatures using reverse Monte Carlo ray tracing (rMCRT). The model is written in CUDA Python, targeting GPU-accelerated computing environments. It models radiative, conductive, and convective heat transfer processes of complex 3D urban environments. A validation against a scale-model outdoor urban experiment is presented.

With many of the new HPC platforms relying on GPUs for much of their computational power, there is a growing need for GPU-accelerated models and tools in the Earth sciences. Solving radiative transfer within urban canopies is for its difficulty to parallelize efficiently, and further advances in models in this area are needed to fully utilize the new HPC platforms in urban climate research.

However, I have some major concerns that need to be addressed before the paper can be reconsidered for publication in GMD.
* * *
**Comments #1**

GUST1.0 as a standalone model has very limited capabilities compared to many other urban surface models. The main novelty is within its radiative transfer modelling, whereas the other parts of the model seem extremely simplistic or even incomplete compared to many other urban surface models (e.g. building-resolving models listed in Table 1 or urban surface models in general as in e.g. Urban-PLUMBER intercomparison by Lipson et al., 2024). The authors do not discuss the scope of applicability and the model's limitations to a necessary extent, nor they adequately justify the publication of a stand-alone model rather than integrating the rMCRT method in a pre-existing model. In my opinion, the current version of the model has too limited real-life applicability for the paper to be considered a substantial contribution.
* * *
**Reply:**

We thank the reviewer for their insightful and constructive feedback, which has helped us clarify the scope, intent, and contribution of this work.

The main contributions of this study are as follows. First, GUST is a building-resolved model with very high spatial resolution (~1 m). Within the Urban-PLUMBER intercomparison, the only other building-resolving model is VTUF-3D. Unlike VTUF-3D (and its prototype TUF-3D), which relies on the radiosity method, our model employs reverse ray tracing and accounts for multiple reflections, making it more suitable for high-density areas with complex building geometries. In contrast, the radiosity method can realistically handle only orthogonal surfaces, which limits its applicability in dense and irregular urban landscapes such as those common in East Asian cities. Second, the model is validated against high-resolution spatiotemporal field measurements and demonstrates excellent accuracy. Third, the GPU-optimized algorithm enables efficient simulation of neighborhood scale while maintaining high spatial and temporal resolution. In response to the reviewer's comments, we have clarified these points in the revised manuscript and explicitly highlighted the main contributions of this study.

In the revised manuscript, we explicitly highlight the main contributions of this study.

This study introduces a GPU-accelerated Urban Surface Temperature model (GUST), which computes radiation using Monte Carlo ray tracing and solves heat conduction with a one-dimensional Monte Carlo random-walk approach. To meet the substantial computational demands of these Monte Carlo simulations, the model employs GPU-based parallel computing for efficient processing. GUST is validated against the high-resolution, scaled outdoor experiment SOMUCH, which provides detailed spatial and temporal measurements.

In the revised manuscript, we emphasize the building-resolved nature of our model to avoid confusion with larger-scale urban land surface schemes.

To tackle urban overheating, a precise understanding of the factors driving excessive surface heat is essential, making accurate modeling of urban surface temperatures a critical step toward developing effective mitigation strategies. Urban surface temperatures are commonly simulated with urban land surface schemes (LSMs). To capture the complex exchanges of energy and momentum within an urban environment, these schemes range from simplified approaches that represent the city as a single impervious slab to advanced frameworks that explicitly incorporate the three-dimensional geometry of buildings with varying heights and material properties. The Urban-PLUMBER project has evaluated 32 such schemes (Grimmond et al., 2010; Grimmond et al., 2011), and classified them into ten categories based on the level of three-dimensional detail represented. The most detailed of these are the building-resolved schemes, which explicitly solve airflow and heat transfer while representing the full three-dimensional urban landscape.

Building-resolved models, such as VTUF (Nice, 2016) and computational fluid dynamics (CFD) tools (Carmeliet and Derome, 2024), solve the governing physical processes at high spatial and temporal resolution. These models are powerful tools for examining the urban thermal balance and identifying the primary drivers of urban heat (Carmeliet and Derome, 2024). They enable a quantitative evaluation of the contribution of each process, such as conduction, radiation, and convection, to the overall thermal balance. This is particularly important for Asia cities, which are characterized by high-density, high-rise developments and complex urban geometry. Findings from the Scaled Outdoor Measurement of Urban Climate and Health (SOMUCH) project highlight the intricate influence of building morphology on the thermal environment, especially under super-high-density conditions (Hang and Chen, 2022). These effects arise from complex three-dimensional urban landscapes, including irregular building forms and intricate shading patterns. Accordingly, models representing high-density Asian cities need greater accuracy and flexibility to account for these features.

In the revised manuscript, we have also expanded the discussion of the model's scope of applicability and limitations to provide a balanced perspective.

This model is a building-resolved urban surface temperature model, focusing on detailed neighborhood-scale processes. Therefore, its application to full city-scale simulations remains limited by computational cost and is currently best suited for neighborhood-scale.

**Comments #2**

The model is not sufficiently described in the paper to allow for, in theory, complete reimplementation of the model by others as required by the GMD policy. The model structure

and numerical methods used to solve the model equations are not sufficiently documented. With respect to the detail required from the model description, I point out the following excerpt from the journal policy:

*"The main paper should describe both the underlying scientific basis and purpose of the model and overview the numerical solutions employed. The scientific goal is reproducibility: ideally, the description should be sufficiently detailed to in principle allow for the re-implementation of the model by others, so all technical details which could substantially affect the numerical output should be described. Any non-peer-reviewed literature on which the publication rests should be either made available on a persistent public archive, with a unique identifier, or uploaded as supplementary information."*

**Reply:**

We thank the reviewer for this critical reminder regarding technical details. In response, we have thoroughly reviewed the manuscript and added more detailed descriptions of the numerical models. To facilitate reimplementation by other researchers, we have also reorganized the code to make it more user-friendly and provided a step-by-step user manual. Furthermore, all model inputs are now consolidated into a single YAML configuration file, which specifies the simulation parameters, weather forcing, geometry paths, surface albedo, and material thermal properties to ensure easy reproducibility.

**Comments #3**

The model for convective heat flux uses a bulk transfer equation, which ignores natural convection. The natural convection becomes especially important in low-wind conditions and whenever the temperature difference between the wall and the atmosphere grows large (see e.g. Fan et al., 2021).

**Reply:**

We thank the reviewer for highlighting this point. In the manuscript, we conducted an energy balance assessment that indicates convective flux plays only a minor role under the SOMUCH measurement conditions. The reviewer's comment rightly notes, however, that this conclusion may not stand for full-scale models.

Two additional factors deserve attention. First, Guangzhou is characterized by generally low wind speeds, which limits the broader applicability of our convective scheme to other cities. Second, the reduced-scale SOMUCH measurements cannot fully represent the natural convective heat transfer that occurs at full scale.

In response, we have added a discussion of convection and potential scale effects in the revised manuscript and included a sensitivity analysis on wind speed to better evaluate the influence of natural convection.

The simulation results were exported in vtk format and visualized using ParaView. Fig. 14 presents the surface temperature distributions at 09:00, 11:00, 13:00, 15:00, 17:00, and 19:00. The movement of building shadows and their influence on surface temperatures are clearly visible in these contours, illustrating the diurnal heating and cooling cycle. These visualizations demonstrate that the model can represent complex building geometries and can be applied to real urban environments.

The energy balance analysis of the SOMUCH experiment indicates that convective heat

transfer plays only a minor role. However, due to the experiment's reduced scale and limited local wind speeds, it remains uncertain whether this conclusion holds at full scale or under higher wind speed conditions.

To further assess the role of the convective model, a wind sensitivity analysis was performed for the real urban configuration. The baseline wind speed (WF = 1.0) was measured on 29 January 2021, the same day used for the validation cases. Wind speeds were then systematically increased by factors of 2.0 and 5.0 relative to the baseline to evaluate their influence on urban surface temperatures. The resulting average surface temperatures of the ground, walls, and roof are shown in Fig. 15. The temperature evolution in Fig. 15 (a)–(c) demonstrates that increasing the wind factor from WF = 1.0 to 5.0 progressively lowers surface temperatures across all urban elements. Fig. 15 (d) quantifies the temperature differences relative to the baseline scenario (WF = 1.0), revealing cooling effects of up to 6 °C, with the most pronounced reductions occurring during peak heating hours. Among the three surfaces, the roof exhibits the greatest sensitivity to wind variations, followed by the ground and then the walls.

These results highlight that, at full scale and under high-wind conditions, convective processes can exert a much stronger influence on urban surface temperatures than indicated by the scaled SOMUCH experiment. Therefore, future studies are needed to better quantify and model convective effects across a broader range of wind speeds and length scales. Moreover, under weak-wind conditions, natural convection becomes especially important, particularly when the temperature difference between the wall and the atmosphere grows large (Fan et al., 2021; Mei and Yuan, 2021). However, this natural-convective effect may not be significant in the scaled SOMUCH experiment.

**Comments #4**

The assumption that the indoor air temperature is equal to the outdoor air temperature is a strong assumption, often invalid. It would require total and efficient ventilation of the indoor air in all conditions, which is not realistic during the heating season or if cooling is applied. This also contradicts the authors' own measurements showing indoor temperatures reaching 40°C (L465)

**Reply:**

We thank the reviewer for this important observation. In the measurements, the building model was enclosed, preventing outdoor air from entering the indoor space. As a result, very high indoor temperatures developed at noon. This is not the case for natural ventilated rooms.

We used simple models for indoor air to minimize model complexity and parameter requirements, allowing us to isolate and focus on the performance of the novel radiative core. The primary focus of this study is to rigorously validate the radiative–conductive–convective coupling model and to highlight the significant performance gains enabled by GPU acceleration. Based on our energy balance analysis, the convective flux plays a minor role, thus his simplification does not compromise the accuracy of the simulated surface temperatures.

We acknowledge, however, that this assumption may introduce uncertainty when indoor air temperatures deviate substantially from outdoor conditions or from the set-point temperature of air conditioning systems. Accordingly, we have clarified the limitations and uncertainties of the indoor air temperature model in the manuscript.

> During the measurements, the building model was enclosed, leading to the development of very high indoor temperatures. Therefore, the measured indoor air temperature was used as an input for the validation simulation.
>
> The dynamic indoor air temperature is not included in this model. It assumes that the indoor air temperature is equal to the outdoor air temperature for a natural ventilated room. This assumption may lead to discrepancies, particularly in situations where indoor temperatures differ from outdoor conditions due to factors such as heat sources, insulation, or limited ventilation.

**Comments #5**

The model setup for the evaluation is not sufficiently documented. Here again, scientific reproducibility should be the goal. I would also recommend presenting some sensitivity analysis with respect to model inputs, or to quantify the uncertainties in another way.

**Reply:**

We thank the reviewer for this valuable suggestion. In the revised manuscript, we have added more detailed documentation of the model setup to enhance scientific reproducibility and reorganized the code to make it more user-friendly. Our original sensitivity analysis focused on the energy balance analysis, but the reviewer rightly noted that the convective heat flux was not thoroughly tested. Because the SOMUCH experiment was conducted with a scaled model and low wind speeds in Guangzhou, we have now included a separate wind speed sensitivity analysis to better quantify the uncertainties associated with convective processes.

> The energy balance analysis of the SOMUCH experiment indicates that convective heat transfer plays only a minor role. However, due to the experiment's reduced scale and limited local wind speeds, it remains uncertain whether this conclusion holds at full scale or under higher wind speed conditions.
>
> To further assess the role of the convective model, a wind sensitivity analysis was performed for the real urban configuration. The baseline wind speed (WF = 1.0) was measured on 29 January 2021, the same day used for the validation cases. Wind speeds were then systematically increased by factors of 2.0 and 5.0 relative to the baseline to evaluate their influence on urban surface temperatures. The resulting average surface temperatures of the ground, walls, and roof are shown on Fig. 15. The temperature evolution in Fig. 15 (a)–(c) demonstrates that increasing the wind factor from WF = 1.0 to 5.0 progressively lowers surface temperatures across all urban elements. Fig. 15 (d) quantifies the temperature

differences relative to the baseline scenario (WF = 1.0), revealing cooling effects of up to 6 °C, with the most pronounced reductions occurring during peak heating hours. Among the three surfaces, the roof exhibits the greatest sensitivity to wind variations, followed by the ground and then the walls.

These results highlight that, at full scale and under high-wind conditions, convective processes can exert a much stronger influence on urban surface temperatures than indicated by the scaled SOMUCH experiment. Therefore, future studies are needed to better quantify and model convective effects across a broader range of wind speeds and length scales. Moreover, under weak-wind conditions, natural convection becomes especially important, particularly when the temperature difference between the wall and the atmosphere grows large (Fan et al., 2021; Mei and Yuan, 2021). However, this natural-convective effect may not be significant in the scaled SOMUCH experiment.

**Comments #6**

Although the authors present evaluation against real-life scale-model measurements, they do not present information on how the model code has been verified. An excerpt from the journal policy:
*"... authors are expected to distinguish between verification (checking that the chosen equations are solved correctly) and evaluation (assessing whether the model is a good representation of the real system). Sufficient verification and evaluation must be included to show that the model is fit for purpose and works as expected."*

**Reply:**

We thank the reviewer for this reminder. We have updated our code and data on Zenodo, which now includes both the full-scale simulation with complex geometry and the reduced-scale validation case. The code has been reorganized for greater user-friendliness, and we have provided a step-by-step user manual. These updates enable users to run and validate the cases independently.

**Comments #7**

Figure 9: The systematic underestimation of west wall temperatures suggests issues with either the radiation model or convective transport. The authors attribute this to sensitivity but don't fully investigate the cause.

**Reply:**

We thank the reviewer for this reminder. We carefully examined the code and performed sensitivity tests. Ultimately, we identified a bug in the shortwave input, which led to the systematic underestimation at west walls. After fixing this bug, the model shows improved performance. We also rewrite the error analysis based on updated results.

Figures 9 and 10 show the comparison of wall temperatures from simulation and measurement. For each surface, multiple points are compared to avoid the influence of

localized anomalies and to ensure that the evaluation reflects the overall wall-temperature behavior. Generally, the wall temperatures are well reproduced, particularly their variation trend. The peak hours are well reproduced. For example, there are two temperature peaks for the west wall. The first one is around 10:00 and the second is around 16:00. Both simulation and measurement show the same occurring time.

To quantify model performance, the coefficient of determination ($R^2$) and root-mean-square error (RMSE) were calculated and marked in each sub-figure. Except for the $H/W = 6$ case, the $R^2$ values exceeded 0.9 for all walls, confirming a strong correlation between simulation and measurement. For $H/W = 6$, $R^2$ is lower because of nighttime underestimation, although the RMSE remains within the same range as the other cases (1.6 °C to 2.2 °C). The main reason for this discrepancy is that wall temperatures in deep street canyons ($H/W = 6$) show only a slight increase compared to the air temperature, due to minimal sunlight penetration into the canyon. Under these conditions, wall temperatures become particularly sensitive to convective and longwave radiative fluxes, which amplifies the impact of small modeling uncertainties.

**Before:**

[Figure]

**After fixing bug:**

[Figure]

**Comments #8**

Although the authors have made the model code public, I cannot find any kind of user manual

for the model. Inclusion of a user manual is required for a model description paper in GMD (see manuscript type policy). Without a user manual it is very difficult for the end users to actually use the model for their applications. A minimal user manual would describe the model installation (or required runtime environment), inputs, outputs, and all other necessary details regarding the model's usage.

**Reply:**

We appreciate the reviewer's suggestion. In response, we have added a detailed user manual "README.md" to the public repository. This manual describes the Python environment setup procedure, specifies the necessary inputs and expected outputs, and provides all other essential details required to run the model.

**Comments #9**

The validation is limited to a single day and a specific experimental setup with uniform materials. Multi-day validation and diverse material properties could strengthen confidence in the results.

**Reply:**

We appreciate the reviewer's valuable comment. To address this point, we have added a comparison over three consecutive days in the Appendix to further demonstrate the robustness of the model. Because the present model is based on SOMUCH measurements with simple concrete surfaces, validation with a wider range of material properties will be pursued in future work.

To further validate the model, we also compared the simulated roof temperatures with measurements over three consecutive days, from 30 January to 1 February 2021, similar to the analysis presented in Fig. 8. The results are shown in Fig. A2, which demonstrates excellent agreement between simulated and observed roof temperatures. By using multiple consecutive days, this comparison minimizes potential bias arising from the single day's weather conditions.

**Comments #10**

I suggest moving Figure 5 and associated analysis into the model evaluation section. After all, analysing the sensitivity of model accuracy with respect to its inputs is part of evaluation. At the same time, I wish the authors would extend the sensitivity analysis to more input parameters.

**Reply:**

We appreciate the reviewer's suggestion. However, the purpose of Figure 5 is not to evaluate model accuracy with respect to input parameters. Rather, it illustrates a key step of the algorithm itself, which is independent of model inputs. Because this figure explains the algorithmic process rather than assessing model performance, we believe it is most appropriate to retain

Figure 5 and its associated discussion in the current section rather than moving it to the model-evaluation section.
* * *
**Comments #11**

Table 1: The authors present an overview of building-resolved models for urban surface temperature. The comparison, however, is rather shallow and does not compare the model features and limitations in depth. I suggest extending the comparison to properly contextualize the new model development.

**Reply**:

We thank the reviewer for the helpful suggestion. In the revised manuscript, we have expanded the discussion around Table 1 to provide a more in-depth comparison of existing building-resolved models, highlighting the limitations of the radiosity method and explaining our rationale for adopting a one-dimensional heat-conduction model and parameterizing convective heat transfer.

The key distinction among these models lies in their radiation schemes, as radiation is the primary energy input into the thermal system of urban surfaces. Moreover, simulating complex urban radiative transfer requires significant computational resources, necessitating simplifications and parameterizations to make the simulation more applicable. For the radiative exchange between urban surfaces, the radiosity method is widely adopted. This approach first collects luminous energy from direct solar and diffuse sky sources and then redistributes reflected energy according to view factors, which quantify the geometric relationships among surfaces. View factors can be determined analytically for simple geometries, estimated with the discrete transfer method (hemisphere discretization and ray counting), or calculated using Monte Carlo ray tracing (MCRT). However, the radiosity method assumes purely diffuse reflections and depends on precise view-factor calculations, making it less accurate for complex urban geometries and surfaces containing semi-transparent materials.

In contrast, the MCRT approach offers greater flexibility and has been widely employed to model solar radiation on complex urban surfaces (Kondo et al., 2001). More recently, its use has expanded beyond radiative transfer to encompass coupled conduction, convection, and radiation processes (Villefranque et al., 2022). In backward MCRT, the energy of the incident light is divided into a large number of photons. By tracking the path of these photons and counting the number of photons absorbed, the net solar radiation reaching a given surface can be calculated. For example, the HTRDR-Urban adopted the backward MCRT, to calculate the solar radiation considering multiple reflections (Schoetter et al., 2023). Building on this concept, Tregan et al. (2023) proposed a theoretical framework to solve linearized transient conduction-radiation problems with Robin's boundary condition in complex 3D urban geometry. Based on that framework, Caliot et al. (2024) developed a probabilistic model to simulate urban surface temperatures, using ray-tracing, walk-on-sphere and double randomization techniques. Their model leverages advancements in computer graphics for image synthesis and the MCM, enabling it to effectively handle large and complex 3D geometries.

The MCRT method has demonstrated strong capability for accurately modeling coupled heat and radiation processes in complex urban environments, but its high computational cost and low efficiency currently limit its application to real-world urban configurations.

**Comments #12**

The authors should clarify the code licensing situation. According to the manuscript, a special collaboration agreement is required to use the code. However, on Zenodo, the license is set to Creative Commons Attribution 4.0 International, which as a public license does not accommodate for such requirements. It is also worth noting that Creative Commons does not recommend their licenses to be used on software (see https://creativecommons.org/faq/#can-i-apply-a-creative-commons-license-to-software). I encourage the authors to investigate other licensing options that are suitable for open publication of the source code. GMD's Code & Data Policy includes some useful information as well.

**Reply:**

We thank the reviewer for this important suggestion. In the revised manuscript, we have removed the statement about requiring a special collaboration agreement and clarified that the code is released under the Creative Commons Attribution 4.0 International license as indicated on Zenodo.

The SOMUCH measurement data are available upon request. The development of GUST, model validation, and visualization in this study were conducted using Python 3.8 with CUDA. The source code, supporting data, and simulation results presented in this paper are archived on Zenodo at https://doi.org/10.5281/zenodo.17138571 and are freely accessible for research purposes under the Creative Commons Attribution 4.0 International (CC BY 4.0) license.

**Comments #13**

L452-453: *"Our previous study has demonstrated that the Monte Carlo ray tracing method has good accuracy in predicting solar radiation."*
The authors should clarify this statement. Either a reference is needed or the statement needs to be justified in the paper.

**Reply:**

We thank the reviewer for the suggestion. In the revised manuscript, we have clarified this statement and added the appropriate reference to our previous study (https://doi.org/10.1016/j.uclim.2025.102363) to support the accuracy of the Monte Carlo ray tracing method in predicting solar radiation.

Figures 12 and 13 show the heat fluxes of walls in the simulation. The heat fluxes of east and west walls are averaged from five measurement points on each. Our previous work (Mei et al., 2025) demonstrated that a Monte Carlo ray-tracing approach accurately predicts incident solar radiation. In that study, we compared the albedo of the urban canopy layer and of street canyons across a range of urban layouts with in-situ measurements, achieving excellent agreement.

**Reply to RC2**

This manuscript presents a GPU-accelerated Urban Surface Temperature model that employs the Monte Carlo method to address complex radiative exchanges and heat transfer processes. The model holds significant potential for various urban applications. It describes the main components of the model, including conduction, solar radiation, longwave radiation, outdoor convection, and an indoor sub-model. The model is validated using field measurements. However, several issues should be addressed in the revised manuscript:

> **Comments #1**
>
> Lack of Justification for Monte Carlo Method: The manuscript does not explain or justify the use of the computationally intensive Monte Carlo method. Urban surfaces are typically characterized by simple geometries, where analytical methods might suffice. A rationale for choosing Monte Carlo over simpler approaches is needed.

**Reply**:

We thank the reviewer for this valuable suggestion. In the revised manuscript, we have added a detailed rationale for selecting the Monte Carlo ray-tracing method. The primary purpose of building this model is to simulate the thermal environment in Asian cities, which are characterized by high-density, high-rise developments and complex urban geometry. Therefore, the SOMUCH experiments focus particularly on high-density urban configurations. The measurements also reveal the intricate influence of building morphology on the thermal environment, particularly under super-high-density conditions. We also emphasize the challenge of complex three-dimensional urban landscapes, including irregular building forms and intricate shading effects, where analytical methods often become less accurate or even impractical. For example, the SUEWS model has exhibited reduced performance in cities such as Shanghai and Singapore. In contrast, the Monte Carlo approach provides the flexibility and accuracy required for these situations, while remaining computationally feasible within our simulation framework.

View factors can be determined analytically for simple geometries, estimated with the discrete transfer method (hemisphere discretization and ray counting), or calculated using Monte Carlo ray tracing (MCRT). However, the radiosity method assumes purely diffuse reflections and depends on precise view-factor calculations, making it less accurate for complex urban geometries and surfaces containing semi-transparent materials.

This is particularly important for Asia cities, which are characterized by high-density, high-rise developments and complex urban geometry. Findings from the Scaled Outdoor Measurement of Urban Climate and Health (SOMUCH) project highlight the intricate influence of building morphology on the thermal environment, especially under super-high-density conditions (Hang and Chen, 2022). These effects arise from complex three-dimensional urban landscapes, including irregular building forms and intricate shading patterns. Accordingly, models representing high-density Asian cities need greater accuracy and flexibility to account for these features.

Limited Scale of Real-World Application: The application to a real urban configuration (p. 27) includes only 40 buildings. This raises concerns about adequacy, as cities typically comprise thousands of buildings. If computational time or hardware limitations restrict the model to such a small scale, its practical applicability may be limited.

**Reply**:

We thank the reviewer for this important suggestion. In the revised manuscript, we specify that the model operates at the neighborhood scale, capturing microscale processes including complex shading patterns, multiple reflections of solar radiation, and longwave radiative exchanges between building surfaces and the ground.

This study aims to develop a GPU-accelerated Urban Surface Temperature (GUST) model to enhance the computational speed of Monte Carlo Method. The model is designed to operate at the neighborhood scale and to capture microscale processes, including complex shading patterns, multiple reflections of solar radiation, and longwave radiative exchanges between building surfaces and the ground. The ultimate objective is to identify the physical drivers of extreme heat in high-density urban neighborhoods.

**Comments #3**

Insufficient Detail in Model Description: Certain aspects of the model, such as the solar radiation sub-model (p. 10), are poorly described. For instance, the manuscript mentions two GPU parallel computing approaches (Fig. 4) but does not clarify what "elements" are, how they are constructed, or how "points" are selected within the domain. Additionally, there is no explanation of how shaded areas are handled, how solar irradiation is calculated over time, or how various urban objects (e.g., buildings, roads, trees, grass, water) are represented in the model.

**Reply**:

We thank the reviewer for the suggestion. In the revised manuscript, we have expanded the Model Description section.

Define computational "elements" and "points":

The GPU parallel computing is executed using two strategies, depending on the total number of elements. In this model, all urban surfaces are represented as triangular facets in STL format, with each triangular facet treated as a single element. Ray tracing and heat-conduction calculations are performed at the centroid of each element. The spatial resolution of the simulation can be refined by using smaller triangular facets, thereby increasing the number of elements. Figure 6 illustrates the triangulated representation of the urban surfaces.

Explain shading treatment:

The solar radiation $q_s$ is calculated on each triangular facet using the reverse Monte Carlo

Ray Tracing (rMCRT) method, which inherently accounts for both shaded and sunlit areas.

Detail temporal calculation:

The solar position is updated at hourly intervals to capture both diurnal and shading variations.

Describe urban object representation:

This model is a building-resolved urban surface temperature model, focusing on detailed neighborhood-scale processes.

The first version focuses on the complex radiative exchange in densely built urban areas.

Glazing and green infrastructure are not included in this experiment.

**Comments #4**

Hardware and Computational Time Details: If the GPU-accelerated Monte Carlo method is suitable for urban surface temperature modeling, the manuscript should provide details on the hardware used for simulations and the computational times for real-world scenarios, starting with the 40-building example.

**Reply**:

We thank the reviewer for the suggestion. In the revised manuscript, we have added information on the hardware specifications and computational time.

Building geometries were imported as STL files comprising approximately $2.3 \times 10^4$ triangular surface meshes. Surface temperatures were calculated on the triangular surface elements, as shown in Fig. 6, with shortwave fluxes resolved by a Monte Carlo ray-tracing scheme using $1 \times 10^5$ photons. The solar position is updated at 30-min intervals to capture both diurnal and shading variations. Transient heat conduction simulations were performed for 24 h with a 10-min time step (600 s) on 29 January 2021, consistent with the validation case. Downward solar radiation, longwave radiation, wind speed, and air temperature were prescribed from the SOMUCH measurements.

The simulation ran on a local workstation with an NVIDIA RTX 5090D GPU and completed in 26.6 h, comprising a view-factor calculation (4.2 h), solar-radiation computation (22.2 h), and coupled heat-transfer analysis (0.2 h).

**Comments #5**

Clarification of Albedo Statement: The statement on lines 43–44, "the complex three-dimensional geometry of urban environments leads to multiple reflections, which reduce urban albedo," requires clarification. Albedo is a material property, and it is unclear how

reflections reduce it. The authors should explain whether this refers to effective albedo or another phenomenon.

**Reply**:

We thank the reviewer for this helpful suggestion. In the revised manuscript, we have clarified that our statement refers to the effective albedo of the urban surface, not the intrinsic material albedo. Multiple reflections within the complex three-dimensional urban geometry increase the probability of radiation absorption, thereby lowering the effective albedo observed at the city scale.

Secondly, the complex three-dimensional geometry of urban environments leads to multiple reflections, which reduce reflected solar radiation and limit the longwave heat loss to sky (Yang and Li, 2015).

**Comments #6**

Figure Improvements:

Fig. 1: Revise to correct typos and include Monte Carlo references in all relevant sub-model descriptions for consistency.
Fig. 2: Correct the typo "Calculatin" to "Calculation."
Fig. 3: Standardize terminology, using either "start point" or "target point" consistently throughout the manuscript.
Terminology Consistency: Clarify the use of "direct," "directional," and "direction" in reference to solar radiation to avoid confusion.
Fig. 5: Specify whether "run time" refers to computational time or the number of model runs.
Fig. 8: Rewrite the figure caption for clarity, as its current wording is difficult to understand.

**Reply**:

We thank the reviewer for the suggestions.

Fig. 1: We have double checked the spelling in Fig. 1 to avoid any typos. The sub-model descriptions are updated for consistency.

Fig. 2: Corrected as suggested.

Fig. 3: We have standardized the terminology throughout the manuscript and now consistently use 'target point,' as well as 'direct' and 'diffuse' for solar radiation.

Fig. 5: We have revised the x-axis label to "Number of runs".

Fig. 8: We have revised the caption to improve clarity.

---

## Author Comment (AC2)

[revised manuscript text omitted]
 Table 1 shows that the radiosity method is widely used to solve the reflections. In the radiosity method, the net longwave and shortwave radiation are solved by two main steps: 1) collecting luminous energy from both the sun and the sky vault, and 2) distributing the reflected energy based on view factors. The luminous energy is influenced by the shading pattern, which is solved by two main approaches in these models: 1) Sunlit shaded distributions method, which employs ray tracing to determine whether a surface is illuminated; and 2) Flux reduction coefficients: where shading is accounted for by reducing the irradiance at shaded points. The reflection and longwave exchange between urban surfaces are determined by view factors, which can be calculated using three approaches: the analytical method, the discrete transfer method, and the Monte Carlo ray tracing method.

- The analytical method uses analytical solutions of view factors by assuming urban surfaces are composed of simple geometries.
- The discrete transfer method (DTM) uses ray tracing method to calculate view factors. The ray direction is determined by dividing the hemisphere into equal segments. This method counts the number of rays intersecting other surfaces.
- The Monte Carlo Ray Tracing (MCRT) is similar to DTM but differs by using rays that are directed randomly. This method is suitable for calculating view factors in complex geometries, but it requires a large number of rays.

The HTRDR-Urban adopted a different approach, using backward MCRT, to calculate the solar radiation considering multiple reflections (Schoetter et al., 2023). The Monte Carlo method (MCM) has been widely used to model solar radiation through the application of a ray tracing algorithm (Kondo et al.,

2001). More recently, its use has expanded beyond radiative transfer to encompass coupled conduction, convection, and radiation processesRecently, its application has been extended to address conduction, convection, and radiation problems (Villefranque et al., 2022). In backward MCRT, the energy of the incident light is divided into a large number of photons. By tracking the path of these photons and counting the number of photons absorbed, the net solar radiation reaching a given surface can be calculated. For example, Tthe HTRDR-Urban adopted a different approach, usingthe backward MCRT, to calculate the solar radiation considering multiple reflections (Schoetter et al., 2023). Building on this concept, Tregan et al. (2023) proposed a theoretical framework to solve linearized transient conductionradiation problems with Robin's boundary condition in complex 3D urban geometry. Based on this that framework, Caliot et al. (2024) developed a probabilistic model to simulate urban surface temperatures, using ray-tracing, walk-on-sphere and double randomization techniques. Their model leverages advancements in computer graphics for image synthesis and the Monte Carlo method (MCM), enabling it to effectively handle large and complex 3D geometries.

The MCRT method has demonstrated strong capability for accurately modeling coupled heat and radiation processes in complex urban environments, but its high computational cost and low efficiency currently limit its application to real-world urban configurations. Although several models listed in Table have been validated against field measurements, others remain unverified and rely on various assumptions and parameterizations, which reduces confidence in their accuracy. Furthermore, the use of field measurement data for model validation faces persistent challenges:The advantage of MCM is its ability to handle complex geometries and albedos, while the disadvantage is its high computational cost.

The low computational efficiency limits the application of MCM in real urban configurations. Although some models in Table 1 are validated against field measurements, others remain unvalidated. These models rely on various assumptions and parameterizations, and the lack of validation limits their accuracy.

Additionally, using field measurement data to validate numerical models faces several 
[revised manuscript text omitted]

The accuracy of wall temperature modeling varies from point to point. There are two main observations from the comparison of wall temperatures.

a) Accuracy Difference Between Walls: The temperatures on the east wall are modeled more accurately than those on the west wall, as the model tends to underestimate the peak temperatures on the west wall.

For $H/W = 1$, the $R^2$ values for west wall temperatures range from 0.95 to 0.98, while those for east wall temperatures range from 0.91 to 0.95. For $H/W = 2$, the $R^2$ values for the west and east wall temperatures show only a slight difference. However, the RMSE values for the west wall, which range from 0.69°C to

1.85°C, are evidently lower than those for the east wall, which range from 0.82°C to 2.53°C. The $R^2$ and

RMSE values for $H/W = 3$ are comparable to those for $H/W = 2$.

[Figure]

[Figure]

[Figure]

(a) $H/W = 1$

(b) $H/W = 2$

(c) *H/W* = 3

(d) *H/W* = 6

[revised manuscript text omitted]

---

## Author Response (AR2)

**Reply to RC1**

We sincerely thank the reviewer for their constructive and insightful comments. The reviewer's original comments are shown in **black**, our detailed responses are provided in **blue**, and the corresponding revisions in the manuscript are highlighted in **grey boxes with red font**.

**Comments #1**

With the revised manuscript, authors have adequately addressed most of the concerns raised by me and the other reviewer in the first review round, but some concerns remain. The authors have substantially improved the description of the scope and intent of their work, but the applicability of the model for real-world scenarios remains very limited.

As the model components beyond the radiative transfer modelling are much more simplistic than those of other neighbourhood-scale models, I think the manuscript adds only limited novelty beyond the authors' earlier work on modelling the urban radiative transfer using MCRT (https://doi.org/10.1016/j.uclim.2025.102363). The work's positioning with one of the journal's main review criteria "Does the manuscript represent a substantial contribution to modelling science within the scope of Geoscientific Model Development (substantial new concepts, ideas, or methods)?" remains my main concern with the manuscript. The situation would be quite different if the other model components beyond the radiative transfer model would be more complete and comparable to those of modern urban surface models.

**Reply:**

We greatly appreciate your feedback, which gives us an excellent opportunity to better highlight the novelty and scientific contribution of our work. The main advances of the present study beyond our earlier MCRT-based urban radiative transfer modelling are as follows:

- **GPU-parallelized framework:** We have developed and implemented a fully GPU-parallelized framework that substantially improves computational efficiency, enabling high-resolution urban simulations at neighbourhood scales that were previously computationally prohibitive. The GPU is utilized not only for the MCRT calculations but also for computing longwave radiative exchange between urban surfaces and solving heat conduction using the Monte Carlo Random Walk method. These algorithms were specifically chosen because they are well suited to the GPU computing framework, allowing for highly parallelized and efficient simulations.

- **High-resolution validation:** The model has been evaluated against detailed observational datasets with both high spatial and temporal resolutions. This comprehensive validation demonstrates the model's ability to accurately capture the fine-scale radiative–convective–conductive heat transfer processes within complex urban configurations.

- **Identification of key physical processes:** Through a detailed surface energy budget analysis, we identify which physical processes are most critical in determining urban energy exchange and how radiative processes interact with conduction and convection components. This insight provides valuable guidance for the future development of urban surface temperature models. For example, our analysis reveals that longwave radiative exchange between urban surfaces plays a critical role, yet this process has often been underrepresented or oversimplified in previous models.

In our revised manuscript, we have more clearly highlighted the main contributions of this study.

To accurately reproduce multiple reflections in high-density urban areas, the radiative heat flux is simulated using a reverse Monte Carlo Ray Tracing method. Sensitivity tests show that $10^5 \sim 10^6$ rays are required for each point to accurately model the solar radiation. This large computational demand for ray tracing is addressed using GPU-based parallel computing. In addition, the GPU is utilized to parallelize both the transient heat conduction, which is solved through random-walk algorithms, and the longwave radiative exchange, which is also computed via ray tracing. This integrated GPU-accelerated framework substantially improves the computational efficiency and scalability of the GUST model.

The comparison with the SOMUCH experiment shows that the transient surface temperatures on roofs, walls and the ground are well reproduced. This comprehensive validation demonstrates the model's ability to accurately capture the fine-scale radiative–convective–conductive heat transfer processes within complex urban configurations. By conducting a surface energy balance analysis, this study demonstrates that longwave radiative exchange between urban surfaces plays a critical role across all building density levels. In contrast, convective heat flux becomes significant only in high-density configurations.

In addition to these methodological advancements, the model's applications extend beyond radiative transfer studies. It can be used for:

- Providing boundary conditions for simulations of the urban outdoor thermal environment and heat-related risk assessment.

- Supporting urban energy consumption analysis, as the simulated surface temperature fields offer critical input for estimating building energy demand and anthropogenic heat release.

- Improving longwave radiation parameterization in mesoscale urban surface models, such as SUEWS or urban canopy models (UCMs).

In our revised manuscript, we have made these contributions more explicit and clarify how the present work contributes to the advancement of urban climate modelling.

Although many additional features will be incorporated into the GUST model in future developments, this does not imply that the current version lacks applicability to real-world scenarios. First, by focusing on the coupled radiative–convective–conductive heat transfer processes, GUST effectively identifies the key physical mechanisms responsible for high urban surface temperatures. Second, it provides high-quality building surface temperature predictions, which can be directly utilized for building energy consumption analyses. Third, the inclusion of longwave radiative exchange between urban surfaces enables GUST to be applied in the parameterization of longwave heat fluxes within mesoscale urban climate models.

**Comments #2**

It also remains unknown for me why the authors have decided not to mention and discuss their earlier published work on using and evaluating the MCRT method for modelling urban radiative transfer, despite being seemingly very much related work. It is mentioned in the

revised manuscript, but merely as a side note in the results section.

**Reply**:

We appreciate the reviewer's comment and the opportunity to clarify this point. Our earlier publication indeed provided the theoretical and algorithmic foundation for the Monte Carlo Radiative Transfer (MCRT) approach, focusing specifically on the parameterization of radiative exchange within the urban canopy layer. However, the present study represents a distinct and independent line of research, with a substantially different objective and model framework.

In this work, we developed an urban surface temperature model that couples radiative, conductive, and convective heat transfer processes within a unified surface energy balance framework. While the MCRT module remains an essential component, the scientific focus has shifted toward understanding and quantifying radiative–convective–conductive interactions and their implications for urban surface temperatures. Therefore, we limited discussion of the earlier MCRT study to avoid redundancy and confusion, as the two works address different scientific questions: the previous one concentrated on methodological developments in urban radiative transfer, whereas the current one emphasizes model coupling and physical process analysis.

In the revised manuscript, we have expanded the references to our previous study to better illustrate the conceptual continuity while maintaining the scientific focus of the current work.

Our previous work (Mei et al., 2025) demonstrated that the MCRT can accurately predict solar radiation in high-density urban configurations, while also achieving high computational efficiency through GPU-based acceleration. In that study, we compared the albedo of the urban canopy layer and of street canyons across a range of urban layouts with in-situ measurements, achieving excellent agreement. The previous study also serves as an independent validation of the ray-tracing component within the modeling framework. Although the ray-tracing procedure in the present study differs from that in our previous work, the core computational framework remains the same. In the previous study, solar rays were emitted directly from the sun and sky, whereas in this study, we adopted a reverse ray-tracing technique, in which rays are emitted from building surfaces toward the surrounding environment.

**Comments #3**

Despite a short extension in the revision, I think the description of the model's limitations still does not clearly communicate the scarcity processes represented in the model compared to more comprehensive urban representations at neighbourhood-scales. Therefore, I think that the statement "This model is a building-resolved urban surface temperature model, focusing on detailed neighborhood-scale processes", especially the usage of the word "detailed" in this context, is not justifiable. I also think the following sentence, "Therefore, its application to full city-scale simulations remains limited by computational cost and is currently best suited for neighborhood-scale", underestimates the model's limitations, although some of them are covered in the following sentences.

**Reply**:

We thank the reviewer for this constructive comment. We also fully understand the concern regarding the necessity of developing a new model when several existing neighbourhood-scale

tools, such as PLAM-4U, ENVI-met, and UrbanMicroClimateFoam, already can simulate urban surface temperature and include many physical processes (e.g., vegetation, glazing, and anthropogenic heat).

The motivation for developing our model is fundamentally different. Rather than aiming to replicate the full complexity of existing urban models, our objective is to systematically validate fundamental physical processes under controlled conditions. The model is built upon the SOMUCH database, which is derived from reduced-scale outdoor measurements in a simplified urban environment. This experimental setup provides an ideal opportunity to isolate and validate individual processes one by one.

For example, the current version demonstrates that the radiative–convective–conductive heat transfer scheme performs very well when compared against detailed SOMUCH observations. In contrast, comprehensive urban models are typically validated only against full-scale urban measurements, where numerous processes occur simultaneously. Such validation approaches often make it difficult to diagnose which sub-models contribute to discrepancies or errors. Therefore, by developing a simplified yet physically explicit model and validating it under controlled, reduced-scale conditions, we can identify which physical processes are most influential and which require further refinement. This process-based understanding is essential for guiding improvements in larger, more comprehensive urban models and for deepening insight into the mechanisms driving the urban heat island effect.

In response to the reviewer's concern, we have clarified our motivation and model positioning in the revised manuscript. We now explicitly state that the purpose of this study is not to compete with or replace existing urban climate models, but to provide a physically consistent, experimentally validated framework that helps to improve the representation of key processes in more complex urban modelling systems.

> The main objective of GUST is to resolve the coupled radiative–convective–conductive heat transfer processes occurring across complex urban geometries. These coupled processes represent one of the core physical mechanisms driving the urban heat island effects (Manoli et al., 2019). The model is developed based on reduced-scale outdoor measurements conducted within a simplified urban environment (Hang and Chen, 2022). In this experimental setup, complex glazing systems and green infrastructure are intentionally excluded to isolate and validate the core radiative–convective–conductive heat transfer mechanisms. GUST uses a time-dependent heat conduction model to couple radiative, convective, and conductive heat transfer processes, as illustrated in Fig. 1.

**Comments #4**

*The introduction has been extended to contextualise the study with relevant prior literature. The authors have also improved the presentation of technical and implementation aspects of the model. The authors did also identify a bug in the model code that caused the spurious wall temperatures. The issue with the lack of user guide has been mitigated. They also provided a small-scale validation case.*

**Reply**:

We thank the reviewer for the valuable feedback provided in this and the previous round, which has greatly helped us to improve the clarity, accuracy, and overall quality of the manuscript.

**Comments #5**

Technical comments:
L52: Asia cities → Asian cities
L146: insensitivity → insensitive
L473: The convective contributes → The convective heat flux contributes
L359: plotted the measurement data → shows the measurement data

**Reply**:

We thank the reviewer for carefully checking the manuscript. All suggested corrections have been implemented.

**Reply to RC2**

**Comments #1**

Please correct the following statement:

L66-67: the complex three-dimensional geometry of urban environments leads to multiple reflections, which reduce reflected solar radiation.

Suggested correction:

The complex three-dimensional geometry of urban environments leads to multiple reflections, which enhance the absorption of solar radiation by surfaces and reduce the net reflected radiation escaping to the atmosphere.*"*

**Reply**:

We thank the reviewer for carefully checking the manuscript. All suggested corrections have been implemented.

Secondly, the complex three-dimensional geometry of urban environments leads to multiple reflections, which enhance the absorption of solar radiation by surfaces and reduce the net reflected radiation escaping to the atmosphere.